# Formulation and Evaluation of Kaempferol Loaded Nanoparticles against Experimentally Induced Hepatocellular Carcinoma: In Vitro and In Vivo Studies

**DOI:** 10.3390/pharmaceutics13122086

**Published:** 2021-12-05

**Authors:** Imran Kazmi, Fahad A. Al-Abbasi, Muhammad Afzal, Hisham N. Altayb, Muhammad Shahid Nadeem, Gaurav Gupta

**Affiliations:** 1Department of Biochemistry, Faculty of Science, King Abdulaziz University, Jeddah 21589, Saudi Arabia; fabbasi@kau.edu.sa (F.A.A.-A.); hdemmahom@kau.edu.sa (H.N.A.); mhalim@kau.edu.sa (M.S.N.); 2Department of Pharmacology, College of Pharmacy, Jouf University, Sakaka 72341, Saudi Arabia; afzalgufran@ju.edu.sa; 3School of Pharmacy, Suresh Gyan Vihar University, Jaipur 302017, India; drgaurav.gupta@mygyanvihar.com; 4Department of Pharmacology, Saveetha Dental College, Saveetha University, Chennai 600077, India

**Keywords:** kollicoat MAE 30 DP/HPMC-AS nanoparticles, hepatoprotective effect, kaempferol, CdCl_2_, Nrf2, Nf-kB

## Abstract

The present study was designed to prepare Kaempferol loaded nanoparticles (KFP-Np) and evaluate hepatoprotective and antioxidant effects in hepatocellular carcinoma models. KFP was encapsulated with hydroxypropyl methylcellulose acetate succinate (HPMC-AS) and Kollicoat MAE 30 DP polymers to prepare nanoparticles (Nps) by quasi-emulsion solvent diffusion technique (QESD). The prepared Nps were evaluated for different pharmaceutical characterization to select the optimum composition for the in vivo assessment. An animal model of cadmium chloride (CdCl_2_)-induced hepatocellular carcinoma in Male Sprague Dawley rats was used in vivo to test the antioxidant and hepatoprotective capacity of free and encapsulated KFP. The prepared Npsshowed nanometric size, low PDI, high drug load as well as encapsulation with a better drug release profile. There was a significant decrease in the increased serum levels of alanine transaminase (ALT), total bilirubin (TBiL), and aspartate transaminase (AST), and the lipid peroxidation’s (MDA) level was attenuated, and levels of markers of the cell antioxidant defence system were restored including Glutathione S-transferase (GST), glutathione (GSH), superoxide dismutase (SOD) and catalase (CAT) via oral pre-treatment with KFP-Np (50 mg/kg b.w. (body weight), 6 weeks). KFP-Np significantly declines an mRNA expression of interleukin-1β (IL-1β), IL-6, and tumor necrosis factor-alpha (TNF-α) as well as decreased nuclear factor kappa-light-chain-enhancer of activated B cells (NF-κB) protein expression. It also upregulated the mRNA expression and protein expression of nuclear factor erythroid 2-related factor 2 (Nrf2) and heme oxygenase-1 (HO-1). While comparing the protective effects of KFP encapsulated in Kollicoat MAE 30 DP and HPMC-AS, Nps was found to be betterthan free KFP. Insummary, result indicate that encapsulation of KFP in NPs provides a potential platform for oxidative stress induce liver injury.

## 1. Introduction

Polyphenolic antioxidant biomolecules have received a lot of attention in recent decades as potential medicinal agents for a variety of diseases like cardiovascular disorders, autoimmune diseases, diabetes, neurodegenerative diseases and cancer. The liver is the body’s most important metabolic organ. It is involved in the digestion, detoxification, preservation, and secretory functions of animals’ bodies. The hepatic disease remains an unsolved global health problem, despite advances in pharmaceutical science. As a result, the scope for modern therapy is still underway [1,2]. Because of their many beneficial effects, such as antioxidant and anti-inflammatory activities, herbal remedies are now one of the most common alternatives for the prevention or treatment of liver disorders. In comparison to other treatments, flavonoids are very effective at treating many liver problems [3]. As a result, intensive study on antioxidant-rich nutritional supplements such as flavonoids and vitamins has been conducted extensively to create more effective therapeutics [4].

Kaempferol (KFP) is a flavonoid present in a wide variety of edible plants like onions, green tea, grapes, potatoes, tomatoes, apples, cucumbers, broccoli, blackberries and green beans [5]. It acts as a scavenger of free radicals and also helps sustain the action of numerous anti-oxidant enzymes like GST, CAT and GSH. KFP exerts its chemo preventive effect via a variety of mechanisms of action such as cell cycle arrest, anti-proliferation, anti-angiogenesis, and induction of apoptosis [6].

Despite its broad range of pharmacological properties, KFP’s usage in biomedical applications is restricted due to its poor water solubility, poor permeability, instability of chemicals in water alkaline medium, extensive metabolic processing before entering the systemic circulation and poor bioavailability at oral administration [7]. It is also a top priority to address these problems, and novel methods have been developed to investigate the flavonoids’ utility in animal models through involving new methods and through optimizing its biological applications, like hepatoprotective effectiveness, via the development of formulations [8,9]. One exciting strategy for addressing these issues is the development of NPs as regulated drug delivery systems for increasing the oral bioavailability of hydrophobic and lipophilic drugs such as KFP [10,11]. This may be done in many ways such as specialized uptake mechanisms, long-term release, longer half-lives (t_1/2_), reduced toxicity, and improved NPs’ capacity to traverse physiological boundaries inside the body of an animal. The study utilizing polymers for the formation of NPs for enhanced oral KFP bioavailability has been conducted [12]. Polymeric Nps have several advantages over traditional drug delivery vehicles, including high durability, the ability to use both hydrophobic and hydrophilic materials, strong resistance to enzymatic degradation of the entrapped drug, the opportunity to reduce the frequency of administration, and, most importantly, minimizing harmful side effects. Nanoformulations may increase the period a medication spends in the body by extending its binding to the GI tract [13]. They will provide a regulated release in a pH-responsive manner whilst still ensuring cell targeting; as a consequence, encapsulated medications have a higher oral bioavailability [14].

Current research aimed to prepare a novel pH-sensitive polymeric nanoparticle prepared with HPMC-AS and Kollicoat MAE 30 DP blend polymers. The prepared Nps were characterized for different parameters. Further, the selected formulation was evaluatedin in vitro cell line study and in vivo hepatocellular carcinoma study on the rat model.

## 2. Materials and Methods

### 2.1. List of Materials

Kollicoat MAE 30 DP (Sigma-Aldrich, St. Louis, MO, USA), Kaempferol (Sigma-Aldrich, St. Louis, MO, USA), HPMC-AS (Sigma-Aldrich, St. Louis, MO, USA), Poloxamer—407 (Loba chemicals, Colaba, Mumbai, India), 1, 2-propylene glycol (Loba chemicals, Colaba, Mumbai, India), perchloric acid, dichloromethane, ethylic ether, methanol and acetonitrile are from Merck, New Jersey, USA.

### 2.2. Preparation of Nanoparticles

Quasi-Emulsion methods were used for the synthesis of eudragit S-100 and HPMC-AS Nps. HPMC-AS and Kollicoat MAE 30 DP in a concentration of 2.5% (*w/v*) was dissolved in methanol and Kaempferol 0.1% (*w/w*) was dissolved in distilled water. Both the solutions were mixed using a syringe pump attached to a Teflon tube. The solution was injected slowly at a rate of 0.25 mL/min into a 50 mLcitrate buffer (pH 5.0) containing poloxamer 1%, *w/v*. The mixture was agitated continuously at 500 rpm during the process of injection. With the solution of polymer-organic solvent in the external aqueous stage and drug-containing pseudo-emulsion in the core step, the mixture is instantly transformed into a biphasic solution. The solution was left at room temperature for 24 h under slow and continuous magnetic stirring. To eliminate polymer aggregations, centrifugation of nano-suspensions was done for 5 min at 1000 rpm. The supernatant, including Nps, was collected and centrifuged for 20 min at 35,000 rpm, and after centrifugation, it was washed three times with distilled water and nanoparticles obtained from the centrifugation were freeze-dried for 24 h [15]. The formulation composition for the preparation of Nps (DNP-1 to DNP-10) of Kaempferol is given in Table 1.

## 3. Evaluation

### 3.1. Particle Characterization

Malvern Instruments Ltd. (DTS Ver.5.10, Serial No. MAL1031371) was used for determining average particle size distribution and polydispersibility index of resulting Kaempferol nanoparticles (KFP-Np). The experiment was carried out at a constant temperature of 25 °C, with transparent disposable zeta cells, and water was used as a dispersant with a viscosity (cP)-0.73 and refractive index (RI)-1.33 [16]. Polydispersibility index is used to describe the distribution of the particle size of NPs. It is considered to be a dimensionless number that ranges between the monodispersed particles value from 0.05 to 0.7 [17]. The Zeta potential of Kaempferol nanoparticles (KFP-Np) was carried out with transparent disposable zeta cells and water was used as a dispersant [18].

### 3.2. Fourier Transformed Infrared (FT-IR) Spectroscopic Analysis

For the verification of the chemical bond’s occurrence between drug and polymer, FT-IR analysis was carried out. For increasing the signal level and reducing the moisture content, clean dry helium gas was carefully purged in detector. Powdered sample was dispersed in KBr powder and pellets were formed by applying pressure of 6000 kg/cm^2^. FT-IR spectrophotometer type 8400S Shimadzu was used for obtaining the FT-IR spectra using powder diffuse reflectance and FT-IR spectrum of Kaempferol and Kaempferol with polymer were compared [19].

### 3.3. Scanning Electron Microscopic (SEM)

Joel SEM analysis instrument, Japan, was used to carry out scanning electron microscopy for exploring external morphology of KFP-Np. Before observation scanning was done at 15 KV accelerating voltage after mounting the samples on an aluminum mount by means of double-sided adhesive tape & vacuum sputtered with gold [20].

### 3.4. Percentage Yield

During the nanoparticle’s preparation the limiting factors for the yield of the product includes quantity of active compound, raw materials, HPMC-AS, kollicoat MAE 30 DP and other process parameters. The yield was calculated by initially measuring the Nps and the percentage yield was determined afterwards with respect to weight of input materials including the weight of the polymers & other additives that were used [21]. For determination of percentage yield, the formula is as follows:
(1)
$$Percentage\;yield=\frac{Weight\;of\;Nps}{\left(Weight\;of\;polymers+Weight\;of\;drug\right)\;}\times 100$$


### 3.5. Encapsulation Efficiency and Drug Loading

A weighted quantity (50 mg) of Nps was suspended in 50 mL of methanol to assess the amount of the substance encapsulated in Nps and sonicated for 15 min in order to thoroughly extract the entrapped drug. The sample was filtered and the resultant solution (1 mL) was withdrawn and diluted up to 50 mLusing phosphate buffer solution. Assay of drug content was done using UV spectrophotometer at 262 nm [22]. The encapsulation efficiency and drug loadingwere calculated using the formula as follows:
(2)
$$\mathrm{EE}\;\left(\%\right)=\frac{Total\;kaempeferol\;added-kaempeferol\;obtained\;after\;sonication\;}{kaempeferol\;added\;}\times 100$$


(3)
$$\mathrm{DL}\;\left(\%\right)=\frac{Total\;kaempeferol\;added-kaempeferol\;obtained\;after\;sonication\;}{Weight\;of\;NPs\;}\times 100$$


### 3.6. In Vitro Drug Release

The dialysis bag was used for studying the in vitro drug release from Kaempferol loaded nanoparticles (KFP-Nps). The study was performed at two different pH for 12 h to check the effect on release behaviour. The two different release media was used 1.2 pH buffer and 6.8 pH phosphate buffer. A dialysis bag (MW cut off 3500 KDa) was used to study the drug release study. The samples were filled in dialysis bag and placed in a beaker containing release media with continuous stirring at a temperature of 37 ± 1 °C. At a specific interval, 1 mLsample was collected and replaced with fresh release media. The released content was evaluated at 262 nm using a UV spectrophotometer [23].

### 3.7. Cell Cultures and Viability

HepG2 (human hepatocarcinoma) cell lines were procured from National Centre for Cell Sciences (NCCS), Pune, India. The cells were cultured in DMEM supplemented with 10% inactivated Fetal Bovine Serum (FBS), penicillin (100 IU/mL), streptomycin (100 μg/mL) and amphotericin B (5 μg/mL) in humidified atmosphere of 5% CO_2_ at 37 °C until confluent. The cells were separated by trypsin phosphate versene glucose (TPVG)solution (0.2% trypsin, 0.02% EDTA, 0.05% glucose in PBS). The stock cultures were grown in 25 cm^2^ culture flasks and all experiments were carried out in 96 microtitre plates (Tarsons India Pvt. Ltd., Kolkata, India).

The cell viabilities of KFP-Np in Cd damaged HepG2 cells were measured by the MTT assay. Briefly, cells were plated at a density of 2 × 10^5^ cells per well in a 96-well flat-bottom microtiter plate at five concentrations (50, 100, 200, 400, 800 μg/mL) of KFP-Np. After a 24 h incubation, the culture media were replaced with media containing CdCl_2_ (20 μM) and incubated for 2 h. At the end of the incubation, 25 μL of MTT solution (5.0 mg/mL) was added to each well and incubated for 4 h at 37 °C. The cells were then lysed with DMSO (200 μL per well), and the reduced intracellular formazan product was quantified in a Bio-Rad enzyme-linked immunosorbent assay microplate reader (680, Bio-Rad, Hercules, CA, USA) at 540nm. Cell viability was presented as the mean of triplicate samples ± SD.

### 3.8. Experimental Procedure for Lab Animals

Central animal house facility of Suresh Gyan Vihar University, Jaipur was used for the animal study.Forty male Sprague Dawley rats (250–270 g; 10–12 weeks of age) were collected from animal house and kept at room temperature (25 °C ± 2), humidity levels (40–70%), and 12 h light-dark period. All animal studies were approved by the institutional animal ethical committee of Suresh Gyan Vihar University, Jaipur (approval no: SGUV/IAEC-7/53).

### 3.9. Experimental Design

The rodents were arbitrarily separated into four test groups of ten rats (*n* = 10), following a week of acclimation. Group 1 (Normal control): received warm (37 °C) isotonic saline (orally, 1 mL/kg daily) for 6 weeks. Group 2 (Cd control): received CdCl_2_ (orally, 10 mg kg^−1^ b.w daily) during the last 4 weeks of the experimental period for induction of hepatotoxicity on the last day. Group 3 (KFP + CdCl_2_): received KFP (orally, 50 mg kg^−1^ b.w daily) for 6 weeks + CdCl_2_ (orally, 10 mg kg^−1^ b.w daily) during the last 4 weeks. Group 4 (KFP-Np + CdCl_2_): received KFP-Np (orally, 50 mg kg^−1^ b.w daily) for 6 weeks + CdCl_2_ (orally, 10 mg kg^−1^ b.w daily) during the last 4 weeks. At the end of this research, all the experimental animals were fully anaesthetized and sacrificed to collect the blood and tissues. The blood sample was collected and centrifuged for 20 min at 3000× *g* at 4° C to separate the serum. Every hepatic tissue was added with 0.15 M KCl of 10-fold solution and homogenize to make up 10% tissue homogenate solution for processing.

### 3.10. Liver Function Markers

The enzyme-mediated processes of AST, ALT and total bilirubin in the serum have been tested by an automated biochemical analyser (Glamour 3000) for diagnosing the hepatic impairments in compliance with the supplier’s directions, using respective kits (NJB Institute, Nanjing, China).

### 3.11. Antioxidant Status in Liver

The actions of superoxide dismutase enzyme, enzyme protecting the cell from oxidative damage like catalase, ligandins isozymes, glutathione, malondialdehyde and total antioxidant capacity (TAC) had been quantified in rat liver according to the procedure of a special kit provided by the supplier (NJB Institute, Nanjing, China). The protein content in the samples was measured as per the Lowry procedure using cow produced serum albumin protein bovine, as a benchmark.

### 3.12. Inflammatory Cytokines

Serum levels of TNF-α, Il-6and Il-1β were measured using a kit of Enzyme-Linked Immunosorbent Assay bought from a supplier (BR Biochem Life Sciences Private Limited, Bhat Bio-Tech, Banglore, India).

### 3.13. Reverse Transcription Polymerase Chain Reaction (RT-PCR)

Total Ribose Nucleic Acid from the hepatocytes was retrieved using Trizol reagent and then reverse transcribed into cDNA using a Prime-Script Reverse Transcription-Polymerase Chain Reaction kit in order to comply with the company’s instructions Fluorescence-ratio polymerase chain reactions were performed on a real-time fluorescence PCR reaction cell as previously stated. Normalizing Gene expression values to β-actin mRNA expression was needed in order to interpret the results. The primer sequences of polymerase chain reactions are as follows [Primer Sequence (5′-3′)]: Nrf2, Forward: ACATGGGTGCTGCGTCG; Reverse: AGTCAAACACCTCTCGC. HO-1, Forward: GTGCCACCAAGTTCAAGCAG; Reverse: CACGCATGGCTCAAAAACCA. Tumor Necrosis Factor-a, Forward: GAGACAGATGTGGGGTGTGA; Reverse: GTCACTCGGGGTTCGAGAAG. Interleukin -6, Forward: CGCTAGCCTCAATGACGACC; Reverse: GGCAGCTTAAAAAGCAACCC. Interleukin -1β, Forward: AGCCATGGCAGAAGTACCTG; Reverse: TGAAGCCCTTGCTGTAGTGG. β actin, Forward: TGAAGCCCTTGCTGTAGTGG; Reverse: TCCAGGGAGGAAGAGGATGC.

### 3.14. Western Blot

The proteins were removed from the freezing hepatocyte by RIPA buffer in which protease and phosphatase inhibitors were mixed and then performed western blot samples. The protein specimen (40 µg) was denatured with 20% SDS solution then diverted to nitrocellulose membrane and blocked with skim milk-based 5% solution then washed and further treated with TBST solution 3 times for 10 min at room temperature. Prime antibodies against Heme oxygenase-1 (1:1000), nuclear Nrf2 (1:1000), p65 nuclear factor-kappa B (1:1000), β-actin (1:5000) and Histone H3 (1:5000) were incubated overnight at 4 °C. Since the sections were washed and blocked with TBST, they were incubated with secondary antibodies again for two hours at a normal room temperature. By using an enhanced chemiluminescence detection kit, the image analysis of protein blots was measured, scanned and analyzed using Image Lab software version 6.0.1.

### 3.15. Histopathology Assay

Tissues were fixed promptly after separation with 10% formaldehyde solution. The liver samples were dehydrated using different concentration levels of ethanol then exposed to xylene and finally embedding in paraffin. These slides were cut at 5 µm thick and then stained with H&E. Histopathological assessments of these cells were done under microscope attached with digital camera.

### 3.16. Data Analysis

The results are shown in the form of mean ± SD. The study was performed with SPSS software (version 19.0). The homogeneity and normality of the different parameter are evaluated by Kolmogorov-Smirnov or Levene test, respectively. Based on statistical analysis, one-way ANOVA were used allowed by Dunnett’s multiple tests to analyze all the results. A value of *p* < 0.05 was found to be important and significant.

## 4. Results

### 4.1. Preparation of Nanoparticles

The pH-sensitive kaempferol Nps were prepared using the modified QESD process. The formulations were prepared with polymers like HPMC-AS and kollicoat MAE 30 DP in different ratio. The optimization is done by selecting the agitation speed, polymer concentration in methanol solution, injection rate and volume of solvent. QESD approach improves the probability of changing particle morphology. Such variables may affect the characterization of Nps.

### 4.2. Characterization

#### 4.2.1. Particle Characterization

Various ratios of kollicoat MAE 30 DP and HPMC-AS were tested to optimize the fabrication procedure of Nps. The mean diameter of formulation DNP-7 was found to be 201 ± 0.45 nm (Figure 1), while the other formulations were found in the range of 248–907 nm.

As known, the P.I. is a parameter used to define the particle size distribution of NPs, it is a dimensionless number and ranges from a value smaller than 0.05 are mainly seen with highly monodisperse standards. The values greater than 0.7 showed that the sample has a very broad particle size distribution. Therefore, it can be stated that the kollicoat MAE 30 DP and HPMC-AS based NPs (DNP-7, DNP-8, DNP-9, DNP-10) prepared by the QESD technique showed homogeneous size distribution (Table 2). The Zeta-potential mainly depends on the chemical nature of the polymer, stabilizing agent, and pH of the medium. Nps prepared from polyester polymers or methacrylate derivative using non-ionic stabilizing agents (poloxamer 407), negative zeta-potential values obtained due to the presence of polymer terminal carboxylic groups. However, it can be stated that the zeta-potential values lower than −10 mV allow good colloidal stability due to the high-energy barrier between particles. (Table 2).

#### 4.2.2. Fourier-Transform Infrared (FT-IR) Spectroscopy

For FT-IR spectroscopic analysis, the Kaempferol (Pure Drug) and optimized formulation were imperilled to compatibility tests to determine if there was some drug and polymers interaction. Kaempferol and DNP-7 formulation’s IR spectra were shown in Figure 2. Kaempferol’s FTIR spectrum showed various characteristic sharp peaks at 1598.50 cm^−1^ (C=C vibration), 1630.68 cm^−1^(C=O Stretching), 2833.30 cm^−1^ (C-H Stretching) and 3218.12 cm^−1^ (corresponding to OH stretching) and the same have been tabulated in Table 3. The optimized formulation’s (DNP-7) spectra displayed all the characteristic absorption bands of Kaempferol with small variations showing the lack of interaction between the drug and the other formulation components.

#### 4.2.3. Scanning Electron Microscopy Analysis (SEM)

SEM technique was used to perform morphological study of the prepared Nps (DNP-7). Figure 3 showed a solid sphere-shaped structure with smooth surface. There were no porous structures or porous surfaces was found; it indicates the rupture of matrix was not an integral constituent in deciding the release profiles.

#### 4.2.4. Percentage Yield

During Nps preparation process, loss of final product is induced by the mechanical variables, and therefore, process yield cannot be 100%. The percentage yield of nanoparticle was calculated by weighing them after drying it and Table 2 shows the obtained data.

#### 4.2.5. Drug Loading/Encapsulation Efficiency

The drug loading and encapsulation efficiency was found to be in range of 11.34% to 15.06% and 30.14% to 46.72%, respectively (Table 2). The lesser entrapment potential can be demonstrated by the drug is slightly soluble in the aqueous phase. So the slightly lesser amount is entrapped in the internal lipid phase, also known as the polymer emulsion period. Drugs that are hydrophilic have a greater concentration in the external water phase; as a result, they have a dispersing propensity in the water phase, while hydrophobic drugs are more likely to stay in the particles.

#### 4.2.6. In Vitro Drug Release

The comparative drug release data of the evaluated Kaempferol nanoparticles were depicted in Figure 4A,B. The drug release study has been evaluated at two different pH, 1.2 and 6.8. The drug release profile showed about 70–90% of Kaempferol release in tested 12 h of study at pH 1.2 (Figure 4A). In initial 2 h, the burst release was achieved (about 20–30% release). The formulation DNP-7 showed the maximum release of 91.98 ± 3.89%. The formulation DNP-1 (68.43 ± 3.44) showed the lowest drug release in same time point.

The drug release study also performed in pH 6.8, and it showed lesser release in comparison to pH 1.2 (Figure 4B). The prepared formulations also showed burst effect in 2 h; the maximum release was found to 15–20%. The maximum release in 2 h was shown by the formulations DNP-6, DNP-7, DNP-8. They depicted Kaempferol release between 15–20%, which is significantly (*p* < 0.05) lower than the release found at pH 1.2. The maximum release was found between 45–60% in the tested 12 h of study. The release was found as follows: 58.98 ± 2.78 (DNP-8)> 54.98 ± 3.65 (DNP-7), 51.28 ± 3.36 (DNP-9). The remaining formulations showed less than 50% release in 12 h.

#### 4.2.7. Effect of KFP-Np on Cell Viability

The CdCl_2_ (20 μM)-exposed HepG2 cells exhibited cell viabilities of 48 ± 3.27 compared to untreated controls (Figure 5). The viability of these CdCl_2_-exposed cells exhibited a dose-dependent effect when pretreated with various KFP-Np concentrations. The percentage viability of KFP-Np + CdCl_2_ at 800 μg/mL dose showed 89 ± 7.83 viability compared to the CdCl_2_-treated control group (Figure 5).

#### 4.2.8. KFP-Np Decreases CdCl_2_-Induced Liver Damage in Rat

The control group had low levels of plasmatic ALT, AST and TBil. However, intragastrically CdCl_2_, at daily 10 mg/kg for four weeks, induced remarkable promotion in the three liver injury markers when compared to normal group (*p* < 0.001). When compared to the model group, the supplement of KFP-Np significantly decreased serum levels of ALT, AST, and TBil (*p* < 0.001). KFP only significantly decreased AST and ALT levels compared to CdCl_2_ treated group (*p* < 0.05) (Table 4).

The antioxidant enzyme activities of SOD, CAT and GST were distinctly decreased (*p* < 0.001) in liver homogenate of CdCl_2_-induced rat when compared to the control group. On the contrary, KFP and KFP-Np treatment significantly ameliorated reductions of these parameters, especially the activities of hepatic SOD, CAT, and GST reached the maximum in KFP-Np, which were significantly higher (*p* < 0.001) than those in the CdCl_2_ group, respectively. Additionally, it was demonstrated that KFP-Np had a 1.6-fold, 1.3-fold and 1.4-fold lower AST, ALT and TBil level whereas 1.3 fold, 1.4 fold and 1.3 fold higher SOD, CAT and GST level compared to KFP. (Table 4).

#### 4.2.9. Histopathologic Examination

Representative images of livers are revealed in Figure 6 and Table 5. The liver cells of the control group displayed normal morphological forms (Figure 6A). In CdCl_2_ group, the extreme liver damages were observed such as hepatic artery dilatation, portal vein obstruction, and congested hepatic vein with focal hepatocyte necrosis (Figure 6B). As a result, KFP-Np-pretreated rats with rat liver damage recover faster. (Figure 6C,D).

#### 4.2.10. Antioxidant Status Evaluated in Rat Liver Tissues

Oxidative stress indicator, malondialdehyde, quickly elevated in response to CdCl_2_ injury by 61.2% as contrasted to normal condition. KFP and KFP-Np substantially improved the CdCl_2_-induced promotion in the hepatic MDA by 27.1, 50.7 percent, respectively (*p* < 0.05 and *p* < 0.001), relative to that in the CdCl_2_ group. Additionally, it was demonstrated that KFP-Np had a 1.4-fold lower hepatic MDA level than KFP. (Figure 7A). The GSH reported a decrease of about 72.1% in the CdCl_2_ group relative to that in the control group. Both groups treated with CdCl_2_ in conjunction with KFP and KFP-Np reported substantial increase in hepatocytes GSH content in comparison with those in the CdCl_2_ group (*p* < 0.05 and *p* < 0.001). Additionally, it was found that KFP-Np exhibited a 1.5-fold increase in hepatic GSH levels when compared to KFP. (Figure 7B). The liver TAC levels in the model group were lower (10.26 ± 1.92 mmol/g tissue protein) than those in the control group (32.19 ± 1.28 mmol/g tissue protein). Administration of KFP and KFP-Np prior to CdCl_2_ showed increased phosphorylation levels of TAC to 23.68 ± 1.21 and 31.27 ± 1.07 mmol/g tissue as compared to CdCl_2_ group. In addition, while comparing KFP-Np to KFP, it was discovered that KFP-Np had a 1.3-fold higher TCA level (Figure 7C). KFP and KFP-Np alone groups showed no changes in MDA, GSH and TCA level compared to control group.

#### 4.2.11. KFP-Np Decreases CdCl_2_-Induced Inflammation in Rat Hepatocytes

It was depicted in Figure 8 that CdCl_2_ resulted in enhanced the expressions of TNF-α, IL-6and IL-1β by around 4.0, 6.0 and 4.0-fold in the liver, and it was revealed that the liver was under inflammatory state. The hepatic mRNA expression of TNF-α, IL-6and IL-1β showed identical patterns when hepatic mRNA expression levels were examined in KFP and KFP-Np treated rats. KFP and KFP-Np pretreatment significantly decreased (*p* < 0.05, *p* < 0.01 and *p* < 0.001) expression levels of TNF-α, IL-6and IL-1β as comparison to the CdCl_2_ group. Additionally, it was found that KFP-Np exhibited a 1.9 fold, 2.9 fold, and 1.5 fold decrease in TNF-α, IL-6and IL-1β when compared to KFP.

#### 4.2.12. KFP-Np Inhibits CdCl_2_-Triggered Pathway of Nrf2 Signaling in Liver

mRNA expression of nuclear Nrf2 and HO-1 was detected and quantified by RTPCR after 4 weeks of CdCl_2_ exposure. CdCl_2_ treatment impaired the down regulation of Nrf2 and HO-1 mRNA expression as comparing with the group of control. Even so, the rates of Nrf2 and HO-1 expression varied between KFP and KFP-Np. KFP and KFP-Np pretreatment upregulated the expression of Nrf2 and HO-1, alike, in Nrf2- and HO-1-KO as contrasted to the CdCl_2_ group. Besides it was also showed that the mRNA expression of nuclear Nrf2 and HO-1 of KFP-Np were 1.5 times greater than KFP. On the contrary, the CdCl_2_ drug reduced substantially the Nrf2 and HO-1 proteins by 51.02 and 52.23% in comparison to the control group, accordingly. The KFP and KFP-Np protein levels showed notable up-regulation against the rats in the CdCl_2_ group. Further it was also found that the Nrf2 and HO-1 proteins levels of KFP-Np were 1.2 and 1.4 fold higher than KFP. KFP and KFP-Np alone groups showed no changes in Nrf2 and HO-1 mRNA and protein expressions compared to control group. (Figure 9A,B).

#### 4.2.13. Inhibits CdCl_2_-Triggered Signaling Pathway NF-κB in Liver Inhibition by KFP-Np

CdCl_2_ caused approximately 3 times greater increase in an NF-κB P65 protein expression in the rat livers in the comparison of normal rat (*p* < 0.001). While, In the KFP and KFP-Np treatment groups, hepatic P65 protein expressions are decreased by 31.94 and 58.89% (*p* < 0.05 and *p* < 0.001) relative to the CdCl_2_ group. Further it was also found that the hepatic P65 protein expressions of KFP-Np were 1.7-fold lower that KFP. KFP and KFP-Np alone groups showed no changes in hepatic P65 protein expressions compared to control group. The finding of NF-κB P65m RNAs was in line with the data of its protein expression. (Figure 10A,B).

## 5. Discussion

The different ratios of HPMC-AS and kollicoat MAE 30 DP were used to prepare kaempferol Nps. The prepared Nps showed the nanometric size particles with the mean diameter of the formulation DNP-7 was found to be 201 nm. The difference in the size was observed due to the variation in the ratio of used polymers. The PDI of NPs ranges between 0.05 to 0.7 and gives narrow range of variability. It can therefore be stated that a homogeneous size distribution was characterized by the HPMC-AS and kollicoat MAE 30 DP based NPs. The pH-sensitive Nps filled with kaempferol were prepared using various pH-sensitive polymers like HPMC-AS and kollicoat MAE 30 DP. The zeta-potential is primarily dependent on the chemical nature of the stabilizing agent, polymer and also on the pH of the medium. The negative values of zeta potential prepared from methacrylate derivatives or polyester polymers use poloxamer 407 as non-ionic stabilizing agents due to the presence of terminal carboxylic groups in polymers. The release profiles of pH-sensitive Kaempferol nanoparticles were tested at various pH including acidic pH (1.2) and physiological pH (6.8). The maximum release from DNP-7 was achieved in pH 1.2 rather than pH 6.8. The initial burst release was also achieved at both pH. The formulation DNP-1 showed the lowest drug release, and DNP-7 showed the maximum drug release in the same time point. The overall conclusion of the study found that the greater solubility was achieved due to amorphization of drug after encapsulating in to NPs [23].

Cd, due to its large region exposure in the atmosphere and nutrients, and longer half could be at a serious threat of liver injury from oxidative stress during an incidence of oxidative stress in this organ. Elevated plasmatic ALT, AST and TBil in rats were exposed to Cd, which is commonly perceived as a direct biomarker for initial hepatic toxicity in several earlier studies. Previous research found that chronic dosing with Cd causes hepatotoxicity [24]. This could be endorsed to the loss and compromise of bilaminar membrane function, this increased the rate at which these biomarkers were released into the circulatory system. Alternatively, different natural antioxidants advocated for the precautionary measures of organ damage caused by cadmium [25]. One of the experimental studies reports that KFP treatment may have therapeutic potential against liver failure caused by several hepatotoxic compounds, such as cisplatin, D-galactosamine, lipopolysaccharide, and valproic acid. Unlike findings in the current report, it was found that oral supplements of KFP and KFP-Np decreased certain inflammation marker levels in liver tissue previously caused by CdCl_2_ [26]. The essential and healing effects of KFP and KFP-Np are demonstrated by the typical hepatic pathological features such as haemorrhage, hepatocyte disarrangement, inflammation, necrosis, and obstruction of a portal vein, which was noticed in CdCl_2_-induced hepatotoxic rats. These negative effects on the hepatic system were partially mitigated in KFP and KFP-Np groups. Thus, the culture of HepG2 cells reveals that CdCl_2_ induces cell death increases as compared to KFP and KFP-Np treated, which increases cell viability [27].

Oxidative damage is an underlying factor in the pathogenesis of Cd-mediated hepatic diseases due to the direct attack of Reactive oxygen species. The increases of intracellular ROS induced the depletion of enzymatic and non-enzymatic antioxidants and accelerate lipid peroxidation, in the case of CdCl_2_ exposure [28]. Current findings showed that after oral administration of CdCl_2_, these levels of antioxidant homeostasis were significantly disrupted. In comparison, the KFP and KFP-Np resulted in significant reductions in the MDA stage, which were followed by both normalizations of hepatic tissue appearance and structure and enhancement of antioxidant ability in the Cd + KFP and KFP-Np groups. These findings also illustrate the alleviating effects of KFP and KFP-Np against CdCl_2_-induced hepatotoxicity; this may be due in part to radical scavenging activity and direct relief of oxidative stress [29].

Nrf2 is considered a defensive transcription factor that could guarantee redox equilibrium by regulating its garget genes that encode a coordinated array of antioxidant enzymes. Under physiological conditions, ubiquitination of Nrf2 is mediated by its interaction with Keap-1 [30]. Under the stimulatory effects of too much ROS and/or electrophiles, Nrf2 breaks away from Keap1, and is transported into the nucleus, where it associates with small Maf proteins to create a heterodimeric complex, then bound to the ARE; subsequently, the more numerous antioxidant and detoxifying enzyme genes are regulated, including HO-1, nitrite oxide or NO-ase, and catalase [31]. CdCl_2_ has contributed to the repression of Nrf2 ubiquitination at Cys151 in a previous analysis. Nrf2 subsequently comes to be relocalized to the nucleus and phosphorylated, which contributes to gene expression of antioxidative functions. The present research proved that the suppression of nucleus Nrf2 in CdCl^2^-administered rat liver tissues indicates the disruption of the nuclear Nrf2 pathway. KFP and KFP-Np react easily with Keap-1 cysteine residues and activating Nrf2 signaling pathway, activating antioxidant enzyme genes transcription. KFP and KFP-Np may have a benefit for the liver against cadmium-induced liver damage via the activation of the Nrf2/ARE signaling pathway [32]. KFP and KFP-Np could activate the up-regulation of Nrf2/ARE pathway that contributes to the oxidative damage caused by CdCl_2_ in testicular tissues and in vitro cultured Leydig cells. The KFP and KFP-Np animals indicate that there are major changes in the nul-Nrf2 and HO-1 levels in the co-administration classes of KFP and KFP-Np + CdCl_2_. It is highly encouraged that KFP and KFP-Np mediated mechanism by which preventive behaviour against Cd-induced oxidative stress in liver tissues is promoted is through Nrf2/HO-1 reinforcement [33].

Additionally, CdCl_2_ was shown to mediate the liver inflammatory response in rats by showing increased inflammatory cytokines TNF-α and IL-6 and IL-1β secretion. In addition, the liver tissues show a similar appearance. This causes chronic inflammation of liver cells that lead to hepatocyte death. Current research indicates that KFP and KFP-Np pretreatment has considerably suppressed NF-κB pathways [34]. The transcription factor NF-κB is involved in the transcription of several pro-inflammatory cytokines, adhesion molecules, and chemokines. Once NF-κB p65 subunit dissociates from IKK and IκB-λ, it then travels to the nucleus to activate transcription of several genes [35]. Present results from animal studies suggest that exposure to CdCl_2_ induces alterations in gene expression of nul-NF-κB signaling. It is important to remember that inflammation and oxidative damage are closely linked during CdCl_2_-mediated liver diseases. Oxidative stress can cause several pro-inflammatory effects by a chain of upstream biological signals; particularly, NF-κB. NF-κB has been recognized as the main Nrf2-mediated upstream/p38 signaling pathway. By elevating Keap1 nuclear translocation, Nrf2/ARE-linked gene expressions abrogated with the activation of NF-κB p65 subunit, competing with Nrf2 for CBP, and limiting CBP-Nrf2 complex formation. It is alleged that its anti-oxidative and anti-inflammatory effects are caused by the inactivation of nuclear factor-κB by KFP and KFP-Np [36,37]. In the current research, the KFP group and KFP-Np group effectively increased and restored the hepatic protein expression of NF-κB inhibitory factor that is inhibited by CdCl_2_. Current findings may have importance on the understanding of the mechanism of KFP and Np-mediated cytoprotective effects against CdCl_2_-induced liver dysfunction by upregulating/Nrf2/ARE pathway and inhibiting NF-κB pathway in addition to the advocating that KFP-Np can alleviate liver inflammation caused by CdCl_2_ by inhibiting NF-kB translocation from the cell membrane to the nucleus.

## 6. Conclusions

The KFP and KFP-Np therapies reduce liver damage through their anti-inflammatory and anti-oxidative functions. The hepatoprotective function of KFP and KFP-Np was influenced at least in part by the modulation of NF-κB/Nrf-2/ARE signaling process in the liver of the rat. It’s the first research to investigate KFP and KFP-Np in Cadmium caused liver failure. Therefore, KFP and KFP-Np administrations should have major consequences for harm prevention from exposure to environmental cadmium. Thus, KFP and KFP-Np administrations could have significant inferences for hepatic injury prevention from exposure to atmospheric and industrial cadmium.

## Figures and Tables

**Figure 1 pharmaceutics-13-02086-f001:**
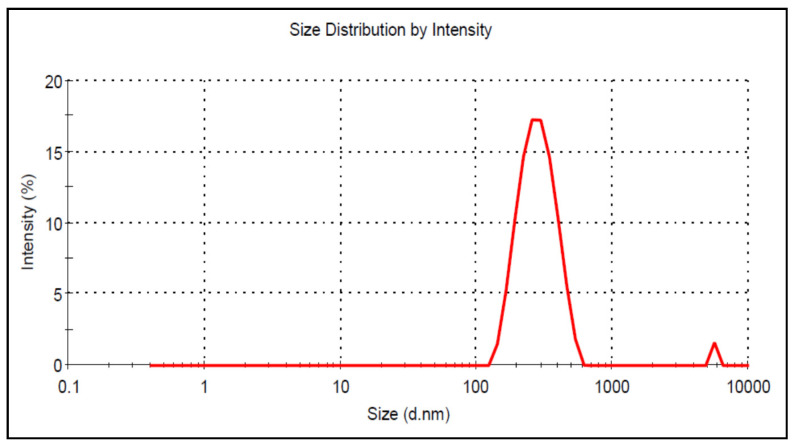
Particle size image of KaempeferolNps (DNP 7).

**Figure 2 pharmaceutics-13-02086-f002:**
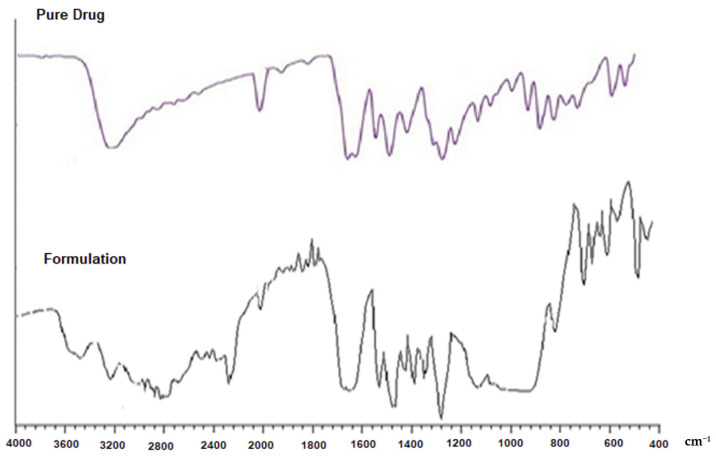
FTIR spectra of the Kaempferol (pure drug) and the prepared Kaempferol nanoparticle.

**Figure 3 pharmaceutics-13-02086-f003:**
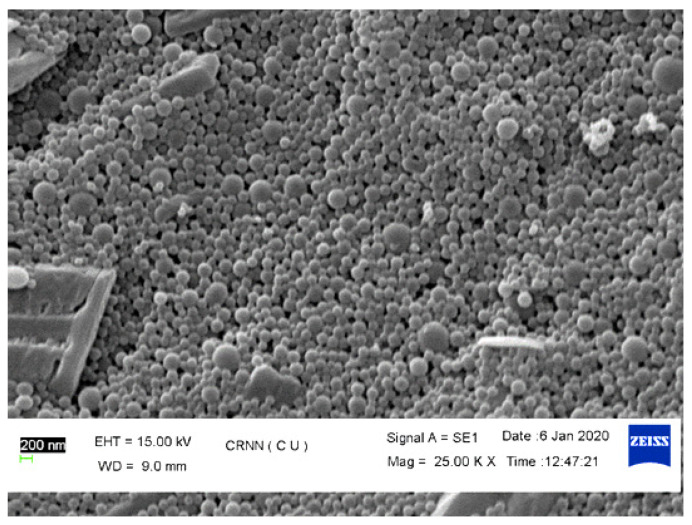
SEM image of Kaempferol nanoparticle (DNP-7).

**Figure 4 pharmaceutics-13-02086-f004:**
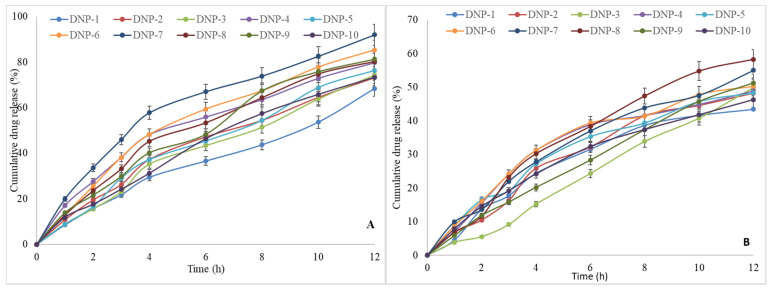
In vitro drug release profile of Kaempferol nanoparticles (DNP-1 to DNP-10) in pH 1.2 (**A**) and pH 6.8 (**B**). Study performed in triplicate and data shown as mean ± SD.

**Figure 5 pharmaceutics-13-02086-f005:**
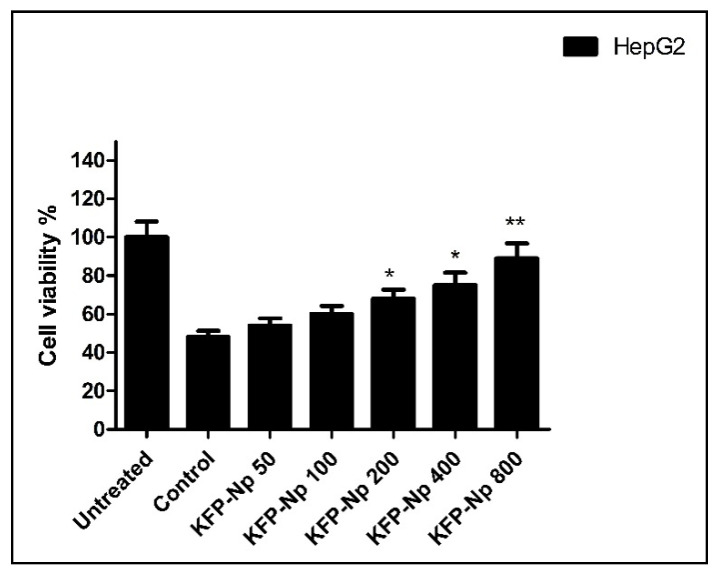
Protective effect of the KFP-Np against CdCl_2_-induced cytotoxicity in a hepatocyte cancer cell line. Untreated, cells alone; Control, cells + CdCl_2_; KFP-Np, cell + CdCl_2_ + KFP-Np. * *p* < 0.05 and ** *p* < 0.01 significantly different from the control group.

**Figure 6 pharmaceutics-13-02086-f006:**
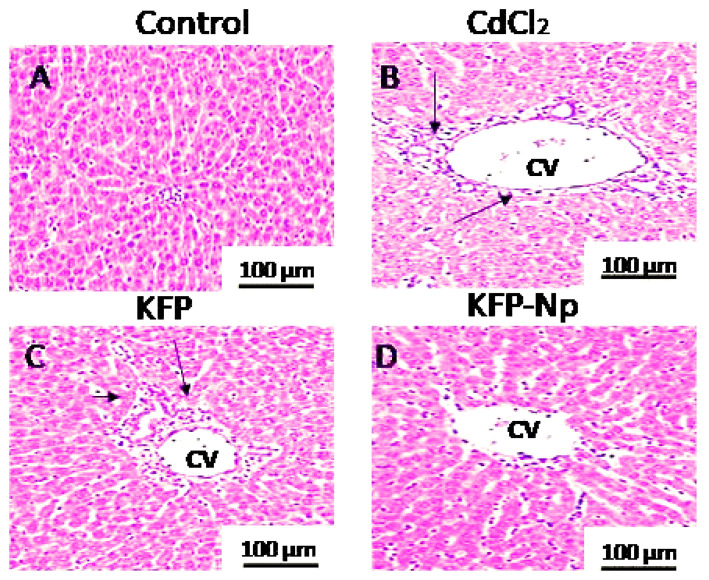
Histological analysis as disclosed by haematoxylin/eosin staining of liver sections from (representative microphotographs). Scale bar: 100µm. (**A**) Control group; (**B**) CdCl_2_-treated group; (**C**) KFP + CdCl_2_; (**D**) KFP-Np + CdCl_2_ (H&E X400). CV (central vein). Black arrow showed aninfilteration of cells.

**Figure 7 pharmaceutics-13-02086-f007:**
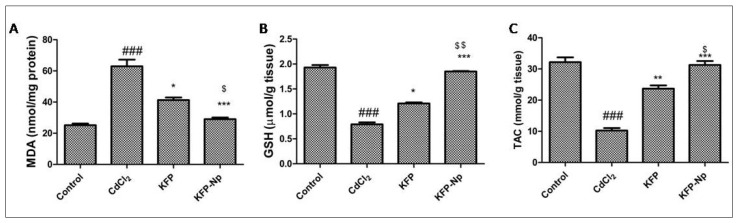
Effects of KFP-Np on CdCl_2_-induced level of MDA, GSH and TAC in liver tissue. (**A**) MDA, (**B**) GSH, (**C**) TAC were measured. Data are presented as means ± SD in each group (*n*= 10). ^###^
*p* < 0.001 significantly different from control group; * *p* < 0.05, ** *p* < 0.01 and *** *p* < 0.001, significantly different from the CdCl_2_ group and ^$^
*p* < 0.05 and ^$$^
*p* < 0.01 significantly different from the KFP group. GSH: reduced glutathione; MDA: malondialdehyde; TAC: total antioxidant capacity.

**Figure 8 pharmaceutics-13-02086-f008:**
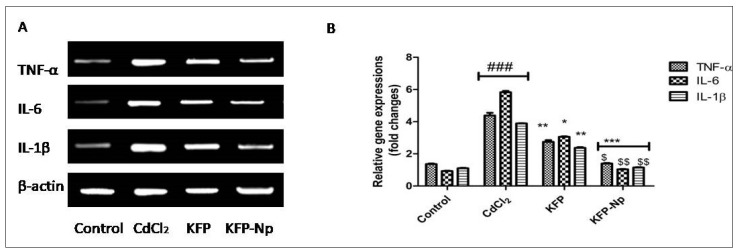
The effects of KFP-Np on inflammation-related gene mRNA expression. Data are presented as mean ± SD in each group (*n* = 10).The relative mRNA levels of TNF-α, IL-6 and IL-1β in different groups were determined by RT-PCR and normalized to the β-actin (**A**). Fold changes of various gene expression in different groups (**B**). ^###^
*p* < 0.001 significantly different from control group; * *p* < 0.05, ** *p* < 0.01 and *** *p* < 0.001, significantly different from the CdCl_2_ group and ^$^
*p* < 0.05 and ^$$^
*p* < 0.01 significantly different from the KFP group.IL-6: interleukin 6; TNF-α: tumor necrosis factor-α; IL-1β: interleukin 1β.

**Figure 9 pharmaceutics-13-02086-f009:**
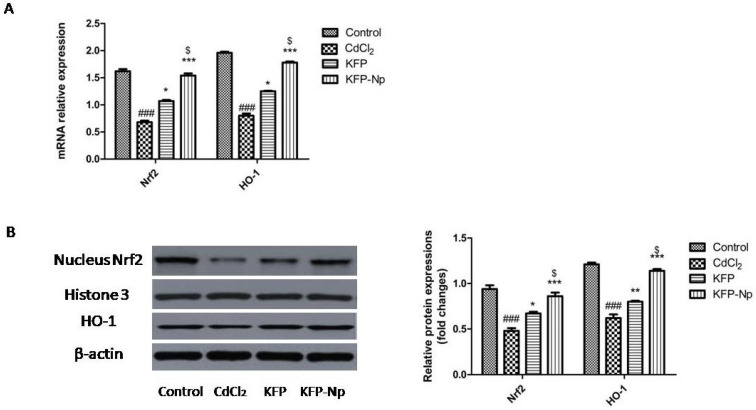
KFP-Np treatment significantly enhanced relative nucleus Nrf2 and HO-1 in liver tissues ofCdCl_2_-induced liver damage rat. The relative mRNA levels of HO-1 and nucleus Nrf2 in different groups were determined by RT-PCR and normalized to the β-actin (**A**). The protein expression levels of HO-1/β-actin and nucleus Nrf2/Histone H3 in different groups were determined by Western blot (**B**). ^###^
*p* < 0.001 significantly different from control group; * *p* < 0.05, ** *p* < 0.01 and *** *p* < 0.001, significantly different from the CdCl_2_ group and ^$^
*p* < 0.05, significantly different from the KFP group. Nrf2: nuclear factor-E2-related factor 2; HO-1: hemeoxygenase-1.

**Figure 10 pharmaceutics-13-02086-f010:**
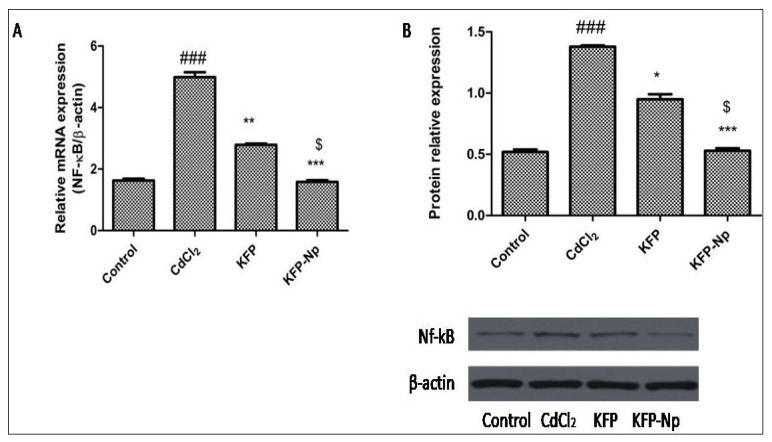
The effects of KFP-Np on nuclear factor-kappa B (NF-κB) (p65) mRNA expression by RT-PCR (**A**), and protein expressions by western blot analysis (**B**). ^###^
*p* < 0.001 significantly different from control group; * *p* < 0.05, ** *p* < 0.01 and *** *p* < 0.001, significantly different from the CdCl_2_ groupand ^$^
*p* < 0.05, significantly different from the KFP group.

**Table 1 pharmaceutics-13-02086-t001:** Formulation chart for the preparation of Kaempferol Nps (DNP-1 to DNP-10).

Formulation	Citrate Buffer (pH 5.0) mL	Poloxamer-407 (%)	Kollicoat MAE 30 DP (%)	HPMC-AS (%)	Drug (mg)
DNP-1	50	1	2	-	100
DNP-2	50	1	1	-	100
DNP-3	50	1	0.5	-	100
DNP-4	50	1	2	0.5	100
DNP-5	50	1	1.75	0.75	100
DNP-6	50	1	1.50	1.0	100
DNP-7	50	1	1.25	1.25	100
DNP-8	50	1	1.0	1.50	100
DNP-9	50	1	0.75	1.75	100
DNP-10	50	1	0.50	2	100

**Table 2 pharmaceutics-13-02086-t002:** Physicochemical data of KaempferolNps formulation. Study has been performed in triplicate and shown as mean ± SD.

Formulation	Particle Size (nm) Mean ± SD	Polydispersity Index	Zeta Potential (mV)	Drug Loading (%)	Encapsulation Efficiency (%)	Percentage Yield (%) ± SD
DNP-1	907 ± 0.51	0.82	−28.5	15.06	39.97	68.2 ± 1.2
DNP-2	778 ± 0.23	0.86	−23.2	12.01	30.14	72.7 ± 1.8
DNP-3	698 ± 0.34	0.88	−20.7	13.01	34.07	69.3 ± 1.7
DNP-4	512 ± 0.54	0.12	−6.6	14.56	41.54	64.7 ± 1.6
DNP-5	499 ± 0.76	0.95	−8.8	13.76	35.23	65.9 ± 1.3
DNP-6	289 ± 0.37	0.81	−10.4	11.34	33.21	61.7 ± 1.2
DNP-7	201 ± 0.45	0.67	−14.8	14.82	46.72	67.3 ± 1.4
DNP-8	248 ± 0.48	0.58	−11.8	14.76	43.5	66.8 ± 1.9
DNP-9	562 ± 0.34	0.21	−7.7	12.45	37.5	65.3 ± 1.3
DNP-10	683 ± 0.14	0.32	−7.5	13.62	32.60	64.9 ± 1.2

**Table 3 pharmaceutics-13-02086-t003:** IR spectra of Kaempferol and DNP-7.

S. No	Kaempferol	Stretching	DNP-7
1	1598.50 cm^−1^	(C=C vibration)	1588.50 cm^−1^
2	1630.68 cm^−1^	(C=O Stretching)	1637.68 cm^−1^
3	2833.30 cm^−1^	(C–H Stretching)	2847.30 cm^−1^
4	3218.12 cm^−1^	OH stretching	3226.12 cm^−1^

**Table 4 pharmaceutics-13-02086-t004:** Effect of KFP-NPs on serum liver function and antioxidant parameters in CdCl_2_ intoxicated rat.

Groups	AST (U/L)	ALT (U/L)	TBil (mg/dL)	SOD (U/L)	CAT (U/L)	GST (mg/dL)
Control	79.36 ± 4.28	38.29 ± 1.27	0.52 ± 0.04	93.28 ± 5.38	62.18 ± 3.05	244.37 ± 19.27
CdCl_2_	185.29 ± 12.71 ^###^	71.28 ± 3.05 ^###^	0.86 ± 0.03 ^###^	48.26 ± 2.19 ^###^	29.43 ± 1.28 ^###^	103.81 ± 8.16 ^###^
KFP + CdCl_2_	139.06 ± 11.29 *	57.44 ± 2.17 *	0.82 ± 0.05	63.21 ± 3.94 *	42.93 ± 2.61 **	173.50 ± 12.17 *
KFP-NP + CdCl_2_	83.28 ± 6.93 ***^$^	41.04 ± 1.52 ***^$^	0.58 ± 0.02 ***^$^	87.67 ± 5.49 ***^$^	59.22 ± 3.29 ***^$^	237.41 ± 19.04 ***^$^

ALT: alanine aminotransferase; AST: aspartate aminotransferase. TBil: total bilirubin. Each value represents the mean ± SD (*n* = 10 in each group); SOD: superoxide diamutase; CAT: catalase; GST: Glutathione S-transferase; each value represents the mean ± SD (*n* = 10). Where ^###^
*p* < 0.001 compared to control group; * *p* < 0.05, ** *p* < 0.01 and *** *p* < 0.001 compared to CdCl_2_ group and ^$^
*p* < 0.05 significantly different from the KFP group.

**Table 5 pharmaceutics-13-02086-t005:** Histopathological alterations of hepatic tissue.

Groups	Inflam	Hem	Nec	HD	CPV
Control	-	-	-	-	-
CdCl_2_	+++	+++	+++	++	+++
KFP + CdCl_2_	+	+	-	+	+
KFP-Np + CdCl_2_	-	-	-	+	-

Histopathological damages including hemorrhage (Hem) in the liver parenchyma, necrosis (Nec), inflammation (Inflam), hepatocyte disarrangement (HD) and congestion of portal vein (CPV) were examined in comparison to control. CdCl_2_, cadmium chloride; KFP, kaempferol; KFP-Np, kaempferol nanoparticle. Damages: - absent, + moderate, ++ severe, +++ very severe.

## Data Availability

Not applicable.

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
