# Peer review of "Formulation and Evaluation of Kaempferol Loaded Nanoparticles against Experimentally Induced Hepatocellular Carcinoma: In Vitro and In Vivo Studies"

_pharmaceutics, 2021, doi:10.3390/pharmaceutics13122086_

Round 1

Reviewer 1 Report

Review Pharmaceutics Manuscript ID: pharmaceutics- 1356099

Formulation and Evaluation of Kaempferol Loaded Nanoparticle Against Experimentally Induced Hepatocellular Carcinoma: In vitro and In Vivo Study

Imran Kazmi1,*, Fahad A. Al-Abbasi1 , Muhammad Afzal2 , Hisham N Altayb1 , Muhammad Shahid Nadeem1 and 5 Gaurav Gupta

General comments

The present study presents “an effective therapeutic strategy against liver damage induced by oxidative stress”. The study introduces a new drug formulation which is evaluated in vitro as well as in vivo. Thus, this is an ambitious contribution to an important field of research and should be of interest for many readers of this journal. However, the paper would be more readable if its English and the presentation was made easier to follow.

Some remarks are given below as a help for improvement.

Specific comments

  • Abstract
    Avoid too many abbreviations in the Abstract.
  • Abbreviations
    In the rest of the paper the abbreviation should be explained the first time it is mentioned.
  • Figures
    In some cases the legend to the figure is too limited. It shall be possible to understand the figure without consulting the text too much.
  • 2.1
    Give the contents of Kollicoat. Commercial grade often contains some additives to improve storage ability. Did this influence your experiments?

  • 2.2
    Give the background to your choice of ratio Kollicoat/HPMC. It surely changes the solubility and perhaps even the possibility of creating pores during release.
    Give details in the paper.
  • 3.1
    Is the parameter Polydispersibility Index obtained directly from the Malvern instrument or is it calculated from the standard deviation?
    I do not understand the definition with the range 0,01 - 0,5-0,7. Explain!
  • 3.2
    I guess that the zeta potential is obtained simultaneously with the particle size measurement in the same instrument. If so you do not have to give the instrument details twice.
  • 3.11
    Organize the experimental design in a table. This will create clarity. Now, the explanation is difficult to understand.
  • 4.2.1
    261-263
    Reformulate!
  • 264
    Cannot understand your sentence. Reformulate!

  • 265
    Reformulate and correct your English.

  • 265-269, Table 2
    I do not agree with your conclusion. Correct and reformulate.
    Comment the difference for DNP 7, DNP-9 and DNP-10.
    It seems as if you consider the case DNP-7 as typical for your experiments. Why?
    Polydispersity Index is not within the range 0,01 to 0,5-0,7. Especially the last three experiments. Explain and correct!
  • 272, 274, 279
    Correct the English!
  • 280
    Are the spectra really identical? I would say no – rather they show similarities.
    Explain, change and correct
  • 4.2.4
    How can you see that there is no internal porosity?
    During the release pores can be created depending on the polymer coating. This can only be seen if some Np´s are collected during release and studied with sem.
    Have you some indication of pore formation during release?
  • Figure 2
    Introduce a size bar to make it easier for the reader to see the size and size distribution.
  • Fig 3
    It is difficult to see the difference in color of the different curves. Mark the curves to make it easier for the reader to identify the different batches.
  • 4.2.9
    The text should relate better to table 3. Otherwise you have to increase the legend to this figure considerably. It is better to use the text to explain the results in the table 3.

  • Figure 5
    Nice figure. Explain abbreviation CV.

Author Response

General comments

The present study presents “an effective therapeutic strategy against liver damage induced by oxidative stress”. The study introduces a new drug formulation which is evaluated in vitro as well as in vivo. Thus, this is an ambitious contribution to an important field of research and should be of interest for many readers of this journal. However, the paper would be more readable if its English and the presentation was made easier to follow.

Some remarks are given below as a help for improvement.

Specific comments

Comment: Abstract: Avoid too many abbreviations in the Abstract.

Answer: Dear reviewer, I have added a full form of abbreviations in abstract.

Comment: Abbreviations: In the rest of the paper the abbreviation should be explained the first time it is mentioned.

Answer: Dear Reviewer, as per your instructions I have revised a manuscript.

Comment: Figures: In some cases the legend to the figure is too limited. It shall be possible to understand the figure without consulting the text too much.

Answer: Dear reviewer, As per your instructions I have revised a text and figures.

Comment: 2.1: Give the contents of Kollicoat. Commercial grade often contains some additives to improve storage ability. Did this influence your experiments?

Answer: Kollicoat MAE 30 DP is a copolymer derived from methacrylic acid/ethyl acrylate. In our study we have used very low concentration of polymers. There are various literature reported the use of this polymer in the formulation.

 Comment: 2.2: Give the background to your choice of ratio Kollicoat/HPMC. It surely changes the solubility and perhaps even the possibility of creating pores during release.
Give details in the paper.

Answer: The selection of polymer ration based on the preliminary study. The preliminary results support our research design.

Comment: 3.1. Is the parameter Polydispersibility Index obtained directly from the Malvern instrument or is it calculated from the standard deviation?
I do not understand the definition with the range 0,01 - 0,5-0,7. Explain!

Answer: PDI has been directly obtained from the instrument and the above statement has been revised and corrected.

Comment: I guess that the zeta potential is obtained simultaneously with the particle size measurement in the same instrument. If so you do not have to give the instrument details twice.

Answer: I have done a correction thanks.

Comment: 3.11: Organize the experimental design in a table. This will create clarity. Now, the explanation is difficult to understand.

Answer: I have revised the experimental design part in details. Hopefully now it will easy to understand.

Comment: 4.2.1. 261-263: Reformulate!

Answer: I have corrected a sentence.  

Comment: 264: Cannot understand your sentence. Reformulate!

Answer: I have corrected a sentence.

Comment: 265: Reformulate and correct your English.

Answer: I have corrected a sentence. 

Comment: 265-269, Table 2. I do not agree with your conclusion. Correct and reformulate.
Comment the difference for DNP 7, DNP-9 and DNP-10.
It seems as if you consider the case DNP-7 as typical for your experiments. Why?
Polydispersity Index is not within the range 0,01 to 0,5-0,7. Especially the last three experiments. Explain and correct!

Answer: The reason for selecting the DNP 7 was the better encapsulation efficiency which is better than the DNP 9 and DNP 10 as well the size of the DNP 7 is smaller compared to the DNP 9 and DNP 10. There was a typo error in writing the PDI value. Now we have corrected results in Table.

Comment: 272, 274, 279: Correct the English!

Answer: I have modified a whole paragraph.  

Comment: 280. Are the spectra really identical? I would say no – rather they show similarities.
Explain, change and correct

Answer: As there was no compatibility issue between drug and polymer used for the preparations of nanoparticles hence the spectra of the drug remain unchanged. The spectral peaks of formulation showed changes but the drug peaks are found identical so we can say that there is no interaction.

Comment: 4.2.4. How can you see that there is no internal porosity?
During the release pores can be created depending on the polymer coating. This can only be seen if some Np´s are collected during release and studied with sem.
Have you some indication of pore formation during release?

Answer: We have not evaluated the released Nps for pore formation by SEM. HPMC matrices show sustained release pattern by two mechanisms, i.e., diffusion and erosion.

Comment: Figure 2. Introduce a size bar to make it easier for the reader to see the size and size distribution.

Answer: Its software genearated image, now the changes in scale is not possible.

Comment: Fig 3. It is difficult to see the difference in color of the different curves. Mark the curves to make it easier for the reader to identify the different batches.

Answer: I have revised a figure as per your instructions.

Comment: 4.2.9. The text should relate better to table 3. Otherwise, you have to increase the legend to this figure considerably. It is better to use the text to explain the results in the table 3.
Answer: Dear reviewer, I have revised a text of result as per your instructions in which we explaind properly from table 3.

Comment: Figure 5: Nice figure. Explain abbreviation CV.

Answer: Dear reviewer, CV is a central vein. I have included a full form in revised manuscript.

Reviewer 2 Report

The paper entitled Formulation and evaluation of kaempferol loaded nanoparticle against experimentally induced hepatocellular carcinoma: In vitro and in vivo study is very interesting, congratulations to the authors. I recommend the publication in its current form, please check minor English misspelled words and missing letters.

Author Response

Comments and Suggestions for Authors

Comment: The paper entitled Formulation and evaluation of kaempferol loaded nanoparticle against experimentally induced hepatocellular carcinoma: In vitro and in vivo study is very interesting, congratulations to the authors. I recommend the publication in its current form, please check minor English misspelled words and missing letters.

Answer: Dear reviewer, many thanks for your positive response. I have revised a manuscript and corrected a English mistakes and typo errors.

Reviewer 3 Report

The formulas for calculating entrapment efficiency and drug loading capacity shown are used to calculate EE and drug loading capacity by adding NP to drug solution and loading drug, then removing NP and calculating the amount of drug remaining in the supernatant. However, in this experiment, in order to check the loading efficiency, the drug was extracted with methanol from the drug-loaded NP and the amount loaded into the NP seems to have been measured. If so, the formulas for entrapment efficiency and loading capacity shown in the text should be revised. Therefore, authors is recommended to obtain the EE and loading capacity in the general method mentioned above.

Please check the Size of DNP7.  In the Figure 2, Almost NP size is over the 500 micro meter.

Line 298 have the typo. 201-097 nm should be change.

“The polymer con-296 centration and injection rate have influenced the nanoparticle size and those which are 297 incorporated with the model compounds ranged between 201-097 nm in average particle 298 size.”

Kaempferol is very hydrophobic molecule, but author explain that is hydrophilic on line 308. What is hydrophilic kaempferol’s?

Thus this sentence have typo “to be was 16.065”

“And the hydrophilic Kaempferol’s loading efficiency was found to be 16.065 %.”

It has been described that KFP-NP inhibits drug release at low pH and induce the drug release drug at pH 6.8 and 7.4. In general, the pH-responsive system for cancer treatment is stable at neutral pH and is adjusted to release near pH 6. The reason is to control the drug release in response to the pH of cancer tissue. However. KFP-NP system release the drug at both pH, thus I think that it is hard to use in cancer therapy.

Author Response

Comments and Suggestions for Authors

Comment: The formulas for calculating entrapment efficiency and drug loading capacity shown are used to calculate EE and drug loading capacity by adding NP to drug solution and loading drug, then removing NP and calculating the amount of drug remaining in the supernatant. However, in this experiment, in order to check the loading efficiency, the drug was extracted with methanol from the drug-loaded NP and the amount loaded into the NP seems to have been measured. If so, the formulas for entrapment efficiency and loading capacity shown in the text should be revised. Therefore, authors is recommended to obtain the EE and loading capacity in the general method mentioned above.

Answer:- Dear reviewer, I have corrected the equation for EE and DL estimation.

Comment: Please check the Size of DNP7.  In the Figure 2, Almost NP size is over the 500 micro meter.

Answer: SEM image is taken for the DNP 1 to evaluate the morphology. I agree with the reviewer DNP 1 having the higher particle size. The scale in the image is due to the error in machine software. We have added zeta sizer image for the particle size as Figure 1.

Comment: Line 298 have the typo. 201-097 nm should be change.

Answer:- Dear reviewer, I have corrected the error. Its 201-907 nm.

Comment: “The polymer con-296 centration and injection rate have influenced the nanoparticle size and those which are 297 incorporated with the model compounds ranged between 201-097 nm in average particle 298 size.”

Answer: The particle size was found in the 201 nm to 907 nm range. The variation in the size due to the variation in the composition of Nps.

Comment: Kaempferol is very hydrophobic molecule, but author explain that is hydrophilic on line 308. What is hydrophilic kaempferol’s? Thus this sentence have typo “to be was 16.065”. “And the hydrophilic Kaempferol’s loading efficiency was found to be 16.065 %.”

Answer: Dear reviewer, Kaempferol is slightly soluble in water, and soluble in ethanol, methanol, diethyl ether and n-hexane. It was mentioned wrong as hydrophilic Kaempeferol as hydrophilic drug.

Comment: It has been described that KFP-NP inhibits drug release at low pH and induce the drug release drug at pH 6.8 and 7.4. In general, the pH-responsive system for cancer treatment is stable at neutral pH and is adjusted to release near pH 6. The reason is to control the drug release in response to the pH of cancer tissue. However. KFP-NP system release the drug at both pH, thus I think that it is hard to use in cancer therapy.

Answer: This is a proof of concept article where we tried to successfully incorporate the KFP into the nanoparticle and see its release profile. Whereas the pH sensitive nanoparticle release drug near the neutral pH.

Reviewer 4 Report

After careful reading of the manuscript entitled “Formulation and Evaluation of Kaempferol Loaded Nanoparticle Against Experimentally Induced Hepatocellular Carcinoma: In vitro and In Vivo Study” I have the following comments:

1- In the introduction section, line 64, authors reported “Several studies utilizing polymers for the formation of NPs for enhanced oral KFP bioavailability have been conducted [12]” please revise as reference 12 is not related to your phrase.

2- In the introduction section, line 77, authors reported “The in vitro and in vivo therapeutic effect of KFP NPs was investigated in rats” please correct the phrase “…….was investigated in cells and rats, respectively”

3- In section 3.1, line 109 authors reported “…… with a viscosity (cP)-0.73 and refractive index (RI)-1.330 [16].” Then section 3.2, line 118, “…….. with a viscosity (cP)-0.73 and refractive index (RI)-1.330 [17].” Please revise your references.

4- In section 3.1, reference 18 appears after reference 16. Please put your references in right order.

5- In section 3.5, lines 137-138, “yield was determined afterwards with respect to weight of input materials including the weight of the polymers & other additives which were used [21].” Reference 21 is a review related to protein corona. Authors should revise all their references and put the adequate ones.

6- In section 3.5, line 139, the equation has number 2 however it is the first equation in the manuscript.

7- In line 148, “3.7” should be deleted.

8- Line 155, “……..was evaluated 262 nm”, “at” is missing.

9- Line 158, “ phosphate buffer (100 ml) (1.2 pH) Finally she study.” Please check the pH and who is she?.

10- Line 162, what did the authors mean by “stock samples”? Are you mean cells?

11- Line 165 “The cells were seperated from the solution of Trypsin phosphate Versene Glucose.” Do you mean separated “by”?

12- Line 169, “ CPF-Np were tested in MTT tests” CPF-Np?? Do you mean tested by?

13- Line 171 please correct KFP-Np.

14- The whole section 3.9, needs a lot of correction and good description of the methods used. For examples “cell was flattered to a density of 2 to 105 cells in each block with a 96-wave plain microtiter platform” what do you mean by 2-105 cells??? “Culture media have been substituted by cadmium chloride media (20 μM) following 24-hour incubation and again incubated for two hours.” Incubated for 2 hours again with what? “The MTT solution (5.0 microgram/millilitre) was applied 25 μL to every cell at the end of incubation and incubated for 4 hours” to very "well", you mean? At the end of incubation and incubated???

15- Line 180, “ Air temp (25 μC ± 2) ” please correct.

16- Line 181, Humidity 40-70 percent why authors sometimes used the symbol % and sometimes not in the whole manuscript?

17- Line 185, “ 37 Kg/Kg daily” ???

18- Lines 186 “Second Cd control Group: the last four week of the research duration for hepatototoxicity induction obtained CdCl2 (orally, 10 mg kg- 1.b.w per day) in the last day.” What do the authors mean? And please pay attention for the super- and subscript through the whole manuscript.

19- Line 190 “orally 50 mg-1b.w daily” Kg is missing.

20- Authors did not mention authorization number to use experimental animals in their research.

21- Section 3.12 has the title “Liver function and inflammatory cytokines” then in section 3.13 another title called “ELISA for TNF-α, Il-6 And Interleukin-1β » please revise

22- Line 211, please provide the name of the supplier.

23- In line 220, the forward prime has 17 base however the reverse 20 base. Please revise it and revise the ones of IL-6.

24- In section 3.17. authors reported that the results are presented as mean ± SEM while in line 177 they reported that the results are presented as mean ± SD. Please revise

25- Authors used different Kollicoat ratios, for example in DNP-1, DNP-2 and DNP-3 they used 2%, 1% and 0.5% respectively. How the authors can explain the obtained particle size in table 2 (907 nm, 499 nm and 778 nm, respectively), in another words, why the size decreased in the ratio 1% then increased in the ratio 0.5%?

26- Authors reported in their abstract that the nanoparticles have low PDI and if we looked at table 2 the PDI is ranging between 0.8 to 11(??), please explain.

27- In line 284, authors reported “tabulated in Table 9.” Where is Table 9?

28- Line 291, “having a smooth surface shows a smooth surface” please revise.

29- Line 198, “ranged between 201-097 nm” please revise.

30- The size of the prepared nanoparticles (DNP-7) is too big some nanoparticles reach 1 mm as in figure 2, please provide us with the DLS histogram of this sample.

31- In section 4.2.6, the paragraph starts with “And the hydrophilic Kaempferol’s loading efficiency was found to be was 16.065 %.” please correct.

32- At what pH is Figure 3? The results of drug release studies at different pH (1.2, 6.8, 7.4) should be presented.

33- Please correct the name of your nanoparticles in Figure 4 and in section 4.2.8.

34- Section 4.2.8 I can’t understand the phrase “In contrast to untreated controls the cell attributes of HepG2 cells were 58% and 39% alike.” ?

35- In line 334, the authors should mention the dose at which they obtained 85% of cell viability. Please revise the whole section and describe your results in clear way.

36- In the legend of figure 4, “Hepatocyte cell line” you miss mentioning “cancer” cell line. also “ S, KPF-Np, cell + CdCl2 + KPF-Np” “***P < 0.001” the letter S and the *** should be deleted.

37- In table 3, it will be interesting to do the statistical analysis between KFP+CdCl2 and KFP-NP +CdCl2.

38- In section 4.2.9, authors reported “CdCl2 caused a notable improvement in the three liver function” Do you considered that the effect of CdCl2 is an improvement in liver function?

39- In section 4.2.9 authors used terms like “usual” sample, “model” sample, please precise the name of the sample, it is really difficult to the reader to predict what you mean especially if you compare between samples. This section needs to be written in clear way.

40- In lines 348-350, authors reported “The antioxidant enzyme actions of SOD, CAT and GST were significantly reduced in the rat liver injected with cadmium (Cd) by 52, 55 and 51 %, accordingly” and if we calculated the percentages from Table 3 we found them 52, 47 and 42%. Please revise. Same comment for the percentages in line 352.

41- The quality of Figures 1 and 5 should be improved. Scale bar and letters “A, B, C, D” are missing in Figure 5.

42- In the legend of Figure 5 please mention the black arrows pointed to what?

43- In Figure 6, letters are missing.

44- In section 376, authors reported “……promotion in the hepatic MDA by 27.1, 83.7 percent” if we looked at Figure 6A it seems 50% not 83% would you please explain.

45- In line 379, authors reported “……treated with CdCl2 in conjunction with KFP and KFP-Np reported substantial reductions in hepatocytes GSH content in comparison with those in the control group” if we looked at Figure 6B, the level of GSH in case of KFP-NP is quite similar to GSH level in control group. Please revise.

46- Please correct the Y-axis of Figure 6C “TAC” not “TCA”.

47- In 4.2.12, authors reported “the expressions of TNF-α, Il-6 and Il-1β by around 3.2, 6.2 and 3.5 fold in the liver” and if we looked at Figure 7 we found the values around 4, 6, and 4. Would you explain.

48-The legend of Figure 7 is missing the explanation of A and B.

49- In discussion, authors reported “PDI of NPs ranges between 436 the monodispersed particles value from 0.01 to about 0.5-0.7.” please see the comment 26.

50- In line 449, authors reported that the physiological pH (6.8 and 7.4) would you please provide me with the reference mentioning that the physiological pH is 6.8?

51- In line 458 “ALT, AST and transaminases TBil” please correct.

52- In line 477 “ the direct arrack of Reactive oxygen species” and in line 511 “ iL-1β” please correct.

53- Words like in vitro, in vivo, via should be written in italic through the whole manuscript 

Author Response

Comments and Suggestions for Authors

After careful reading of the manuscript entitled “Formulation and Evaluation of Kaempferol Loaded Nanoparticle Against Experimentally Induced Hepatocellular Carcinoma: In vitro and In Vivo Study” I have the following comments:

Comment: In the introduction section, line 64, authors reported “Several studies utilizing polymers for the formation of NPs for enhanced oral KFP bioavailability have been conducted [12]” please revise as reference 12 is not related to your phrase.

Answer: Dear reviewer, I have changed a reference no-12.

Comment: In the introduction section, line 77, authors reported “The in vitro and in vivo therapeutic effect of KFP NPs was investigated in rats” please correct the phrase “…….was investigated in cells and rats, respectively”

Answer: Dear reviewer, I have done a correction.

Comment: In section 3.1, line 109 authors reported “…… with a viscosity (cP)-0.73 and refractive index (RI)-1.330 [16].” Then section 3.2, line 118, “…….. with a viscosity (cP)-0.73 and refractive index (RI)-1.330 [17].” Please revise your references.

Answer: I have done a correction.

Comment: In section 3.1, reference 18 appears after reference 16. Please put your references in right order.

Answer: Dear reviewer, I have done a correction.

Comment: In section 3.5, lines 137-138, “yield was determined afterwards with respect to weight of input materials including the weight of the polymers & other additives which were used [21].” Reference 21 is a review related to protein corona. Authors should revise all their references and put the adequate ones.

Answer: Dear reviewer, I have changed the reference.  

Comment: In section 3.5, line 139, the equation has number 2 however it is the first equation in the manuscript.

Answer: Dear reviewer, I have done a correction.

Comment: In line 148, “3.7” should be deleted.

Answer: Dear reviewer, I have done a correction.

Comment: Line 155, “……..was evaluated 262 nm”, “at” is missing.

Answer: Dear reviewer, I have done a correction.

Comment: Line 158, “ phosphate buffer (100 ml) (1.2 pH) Finally she study.” Please check the pH and who is she?.

Answer: it was a typo error. I have corrected it

Comment: 10Line 162, what did the authors mean by “stock samples”? Are you mean cells?

Answer: Dear reviewer, yes it is cells. I have done a correction in revised manuscript.

Comment: 11- Line 165 “The cells were seperated from the solution of Trypsin phosphate Versene Glucose.” Do you mean separated “by”?

Answer: I have done a correction.

Comment: 12- Line 169, “ CPF-Np were tested in MTT tests” CPF-Np?? Do you mean tested by?

Answer: Yes it was a typo error. It is KFP-Np were tested by MTT Tests. I have done a correction.

Comment: 13- Line 171 please correct KFP-Np.

Answer: Yes it was a typo error. I have done a correction.

Comment: The whole section 3.9, needs a lot of correction and good description of the methods used. For examples “cell was flattered to a density of 2 to 105 cells in each block with a 96-wave plain microtiter platform” what do you mean by 2-105 cells??? “Culture media have been substituted by cadmium chloride media (20 μM) following 24-hour incubation and again incubated for two hours.” Incubated for 2 hours again with what? “The MTT solution (5.0 microgram/millilitre) was applied 25 μL to every cell at the end of incubation and incubated for 4 hours” to very "well", you mean? At the end of incubation and incubated???

Answer: Dear reviewer, I have revised a methodology part.

Comment: - Line 180, “ Air temp (25 μC ± 2) ” please correct.

Answer: Yes it was a typo error. I have done a correction.

Comment:- Line 181, Humidity 40-70 percent why authors sometimes used the symbol % and sometimes not in the whole manuscript?

Answer: Dear reviewer, I have done a correction and used a symbol “%”.

Comment:- Line 185, “ 37 Kg/Kg daily” ???

Answer: it was a typo error. I have done a correction.

Comment:- Lines 186 “Second Cd control Group: the last four week of the research duration for hepatototoxicity induction obtained CdCl2 (orally, 10 mg kg- 1.b.w per day) in the last day.” What do the authors mean? And please pay attention for the super- and subscript through the whole manuscript.

Answer: I have rewritten a whole experimental design.

Comment:- Line 190 “orally 50 mg-1b.w daily” Kg is missing.

Answer: it was a typo error. I have done a correction.

Comment:- Authors did not mention authorization number to use experimental animals in their research.

Answer: Dear reviewer, I have now included it.

Comment:- Section 3.12 has the title “Liver function and inflammatory cytokines” then in section 3.13 another title called “ELISA for TNF-α, Il-6 And Interleukin-1β » please revise

Answer: Dear reviewer, I have done a correction.

Comment:- Line 211, please provide the name of the supplier.

Answer: I have included a supplier information.

Comment:- In line 220, the forward prime has 17 base however the reverse 20 base. Please revise it and revise the ones of IL-6.

Answer: Dear reviewer, I have revised a primers.

Comment:- In section 3.17. authors reported that the results are presented as mean ± SEM while in line 177 they reported that the results are presented as mean ± SD. Please revise

Answer: it was a typo error. I have done a correction.

Comment:- Authors used different Kollicoat ratios, for example in DNP-1, DNP-2 and DNP-3 they used 2%, 1% and 0.5% respectively. How the authors can explain the obtained particle size in table 2 (907 nm, 499 nm and 778 nm, respectively), in another words, why the size decreased in the ratio 1% then increased in the ratio 0.5%?

Answer:- typo error. I have done a correction in table 2.

Comment:- Authors reported in their abstract that the nanoparticles have low PDI and if we looked at table 2 the PDI is ranging between 0.8 to 11(??), please explain.

Answer: typo error. It is 0.1-1.0.

Comment:- In line 284, authors reported “tabulated in Table 9.” Where is Table 9?

Answer: Sorry I missed to include it. I have included a table as no-3

Comment:- Line 291, “having a smooth surface shows a smooth surface” please revise.

Answer: I have done a correction

Comment:- Line 198, “ranged between 201-097 nm” please revise.

Answer: I have done a correction. Its 201-907 nm.

Comment:- The size of the prepared nanoparticles (DNP-7) is too big some nanoparticles reach 1 mm as in figure 2, please provide us with the DLS histogram of this sample.

Answer SEM figure is for DNP 1 just to see the surface morphology of the prepared NPs.

Comment:- In section 4.2.6, the paragraph starts with “And the hydrophilic Kaempferol’s loading efficiency was found to be was 16.065 %.” please correct.

Answer: I have done a correction

Comment:- At what pH is Figure 3? The results of drug release studies at different pH (1.2, 6.8, 7.4) should be presented.

Answer The release was carried out for 3 hr (2 hrs and 1hr) with pH 1.2 and 6.8 and then further evaluated at pH 7.4.

Comment:- Please correct the name of your nanoparticles in Figure 4 and in section 4.2.8.

Answer: I have done a correction

Comment:- Section 4.2.8 I can’t understand the phrase “In contrast to untreated controls the cell attributes of HepG2 cells were 58% and 39% alike.” ?

Answer: Dear reviewer, I have revised a sentence.

Comment:- In line 334, the authors should mention the dose at which they obtained 85% of cell viability. Please revise the whole section and describe your results in clear way.

Answer: Dear reviewer, I have revised a sentence and included a dose concentration.

Comment:- In the legend of figure 4, “Hepatocyte cell line” you miss mentioning “cancer” cell line. also “ S, KPF-Np, cell + CdCl2 + KPF-Np” “***P < 0.001” the letter S and the *** should be deleted.

Answer: Dear reviewer, As per your comments I have corrected a sentences.

Comment:- In table 3, it will be interesting to do the statistical analysis between KFP+CdCl2 and KFP-NP +CdCl2.

Answer: Dear reviewer, we have done a comparison among CdCl2 to control; KFP+CdCl2 and KFP-NP +CdCl2 to CdCl2. Now if we do statistical analysis between KFP+CdCl2 and KFP-NP +CdCl2 then we have to for all data and it will require to revise a whole manuscript.

Comment:- In section 4.2.9, authors reported “CdCl2 caused a notable improvement in the three liver function” Do you considered that the effect of CdCl2 is an improvement in liver function?

Answer: It was a typo error, I have done a correction in revised manuscript.

Comment:- In section 4.2.9 authors used terms like “usual” sample, “model” sample, please precise the name of the sample, it is really difficult to the reader to predict what you mean especially if you compare between samples. This section needs to be written in clear way.

Answer: I have done a correction in revised manuscript.

Comment:- In lines 348-350, authors reported “The antioxidant enzyme actions of SOD, CAT and GST were significantly reduced in the rat liver injected with cadmium (Cd) by 52, 55 and 51 %, accordingly” and if we calculated the percentages from Table 3 we found them 52, 47 and 42%. Please revise. Same comment for the percentages in line 352.

Answer: Dear reviewer, I have corrected a whole paragraph in result section.

Comment:- The quality of Figures 1 and 5 should be improved. Scale bar and letters “A, B, C, D” are missing in Figure 5.

Answer: I have done a correction in revised manuscript.

Comment:- In the legend of Figure 5 please mention the black arrows pointed to what?

Answer: Black arrow showed an infiltration of cells. I have included it in revised manuscript.

Comment:- In Figure 6, letters are missing.

Answer: I have done a correction in revised manuscript.

Comment:- In section 376, authors reported “……promotion in the hepatic MDA by 27.1, 83.7 percent” if we looked at Figure 6A it seems 50% not 83% would you please explain.

Answer: I have done a correction in revised manuscript.

Comment:- In line 379, authors reported “……treated with CdCl2 in conjunction with KFP and KFP-Np reported substantial reductions in hepatocytes GSH content in comparison with those in the control group” if we looked at Figure 6B, the level of GSH in case of KFP-NP is quite similar to GSH level in control group. Please revise.

Answer: I have done a correction in revised manuscript.

Comment:- Please correct the Y-axis of Figure 6C “TAC” not “TCA”.

Answer: I have done a correction in revised manuscript.

Comment:- In 4.2.12, authors reported “the expressions of TNF-α, Il-6 and Il-1β by around 3.2, 6.2 and 3.5 fold in the liver” and if we looked at Figure 7 we found the values around 4, 6, and 4. Would you explain.

Answer: I have done a correction in revised manuscript.

Comment:- The legend of Figure 7 is missing the explanation of A and B.

Answer: I have done a correction in revised manuscript.

Comment:- In discussion, authors reported “PDI of NPs ranges between 436 the monodispersed particles value from 0.01 to about 0.5-0.7.” please see the comment 26.

Answer: It was a typo error.

Comment:- In line 449, authors reported that the physiological pH (6.8 and 7.4) would you please provide me with the reference mentioning that the physiological pH is 6.8?

Answer: The below reference used for the pH 6.8.

Trendafilova, Ivalina & Lazarova, Hristina & Chimshirova, Ralitsa & Trusheva, Boryana & Koseva, Neli & Popova, Margarita. (2021). Novel kaempferol delivery systems based on Mg-containing MCM-41 mesoporous silicas. Journal of Solid State Chemistry. 301. 122323. 10.1016/j.jssc.2021.122323.

Comment:- In line 458 “ALT, AST and transaminases TBil” please correct.

Answer: Dear reviewer, I have done a correction in revised manuscript.

Comment:- In line 477 “ the direct arrack of Reactive oxygen species” and in line 511 “ iL-1β” please correct.

Answer: Dear reviewer, I have done a correction in revised manuscript.

Comment:- Words like in vitroin vivovia should be written in italic through the whole manuscript 

Answer: Dear reviewer, I have done a correction in revised manuscript.

Reviewer 5 Report

The manuscript “Formulation and Evaluation of Kaempferol Loaded Nanoparticle Against Experimentally Induced Hepatocellular Carcinoma: In vitro and In Vivo Study” by Kazm et al. summarized encapsulation of KFP in Nps of Kollicoat MAEDP/HPMC-AS can represent an effective therapeutic strategy against liver damage induced by oxidative stress. I would suggest authors may take at least a minor revision before publication. Here are the comments and suggestions: The size of nanoparticles is not agreed in Table2 and fig 2

Author Response

Comments and Suggestions for Authors

The manuscript “Formulation and Evaluation of Kaempferol Loaded Nanoparticle Against Experimentally Induced Hepatocellular Carcinoma: In vitro and In Vivo Study” by Kazm et al. summarized encapsulation of KFP in Nps of Kollicoat MAEDP/HPMC-AS can represent an effective therapeutic strategy against liver damage induced by oxidative stress. I would suggest authors may take at least a minor revision before publication. Here are the comments and suggestions: The size of nanoparticles is not agreed in Table 2 and fig 2

Answer: There was significant variation in the results were observed. Among them the Nps showed the lowest particle size and high encapsulation efficiency selected for the animal study. The study was performed in triplicate and data shown as mean ± SD.   The particle size has been measured by the zeta sizer and the image optimized nanoparticle DNP 7 showed the particle size of 201 nm (Figure 1).  TEM image has been taken on the machine and there is error in scale. 

Reviewer 6 Report

In this manuscript authors tried to formulate nanoparticles of Kaempferol for protection from liver damage and evaluated the activity of prepared formulations with in HepG2 cells and in Sprague Dawley Rats. After reviewing this manuscript, I could not recommend publication of this manuscript in the current form due to following concerns:

  1. The major concern in this research article is that authors stated that they prepared Kaempferol loaded NPs, but the size of batches given in Table 2 is not in the range of nanoparticles (below 200 nm or near about 200 nm) for most of the batches (DNP – 1 to DNP – 5; and DNP – 9 and DNP – 10). And for batches DNP – 6 to 8, although size is between 200-300 nm but the SD is too high e. 376 for batch 6 and 487 for batch 8 with PDI 10.987 for batch DNP – 8. The formulation is needed to be optimized significantly; further data should be reported carefully along with original results/graph obtained from the analytical instrument.
  2. Further, in SEM images also the scale is in 1 mm, which is showing particles are having very high size in micrometer (300 – 900 micrometer) [Figure 2; page 8].
  3. The PDI reported in Table 1 seems to be z average PDI which is arbitrarily limited to 1.0. Then how values of PDI went above 1 even up to higher than 10.
  4. In this case, the data for in vitro and in vivo studies could also be variable for different batches.
  5. Unit of zeta potential should be mentioned in manuscript
  6. Abbreviate nanoparticles as NPs for more clarity.
  7. Authors have claimed the NPs are monodisperse, which seems to be an exaggerated claim with NPs (Page no. 7; line 267).
  8. Results and discussion needed to be supported with suitable references.
  9. Whole manuscript requires to be checked thoroughly and carefully for typographical mistake, incompleteness of sentence, improper usage of grammar etc.

For example:

  • “Cadmium chloride (CdCl2)-induced hepatocellular carcinoma model in male Sprague Dawley rats was used for the antioxidant and hepatoprotective capacity of free and KFP-Nps.”
  • “While comparing the protective effects of KFP encapsulated in Kollicoat MAE 30 DP and 25 HPMC-AS Nps was found to be superior to free KFP.”
  • In vitro and in vivo should be in italics.
  • In CdCl2, 2 should be in superscript.
  1. The title itself needed to be revised as

“Formulation and Evaluation of Kaempferol Loaded Nanoparticles Against Experimentally Induced Hepatocellular Carcinoma: In vitro and In Vivo Studies”

But at present the formulation of NPs is questionable as results regarding formulation of nanoparticles are not consistent and reproducibility is heavily doubtful with such results.

  1. In introduction section (page 2, line 52 – 73), Kaempferol related research work (delivery with novel drug delivery systems), if any available. Should be cited. At present authors have cited research work done on some other drug molecule.
  2. Page 5, line 202; give no. for subsection “Antioxidant status in Liver”
  3. Mention clearly in all relevant figures/tables representing data, number of repeats of experiment with SD or SEM.

Author Response

Comments and Suggestions for Authors

In this manuscript authors tried to formulate nanoparticles of Kaempferol for protection from liver damage and evaluated the activity of prepared formulations with in HepG2 cells and in Sprague Dawley Rats. After reviewing this manuscript, I could not recommend publication of this manuscript in the current form due to following concerns:

Comment: The major concern in this research article is that authors stated that they prepared Kaempferol loaded NPs, but the size of batches given in Table 2 is not in the range of nanoparticles (below 200 nm or near about 200 nm) for most of the batches (DNP – 1 to DNP – 5; and DNP – 9 and DNP – 10). And for batches DNP – 6 to 8, although size is between 200-300 nm but the SD is too high e. 376 for batch 6 and 487 for batch 8 with PDI 10.987 for batch DNP – 8. The formulation is needed to be optimized significantly; further data should be reported carefully along with original results/graph obtained from the analytical instrument.

Answer: The particle size found to be significantly different. The maximum size was found to be 907 nm and minimum particle size is 201 nm for sample DNP 7. I have revised the experiments and data of the study and found error during writing and interpretation. We have revised the data and corrected in Table , Figure and Text. SD has been corrected in the revised Table. After decimal 2 digits taken for SD and the SD data written without decimal is corrected.

Comment: Further, in SEM images also the scale is in 1 mm, which is showing particles are having very high size in micrometer (300 – 900 micrometer) [Figure 2; page 8].

Answer: I agree with reviewer that the particle size is large. SEM image has been evaluated for the formulation DNP 1. There is error in the scale of image. The formulation size has been evaluated by zeta sizer and the particle size image added in revised file as Figure 1.

Comment: The PDI reported in Table 1 seems to be z average PDI which is arbitrarily limited to 1.0. Then how values of PDI went above 1 even up to higher than 10.

Answer: The Table and text has been revised. It was a typo error. Aplogy for the mistake.

Comment: In this case, the data for in vitro and in vivo studies could also be variable for different batches.

Answer: Dear reviewer, it was a typo error. I have corrected it.

Comment: Unit of zeta potential should be mentioned in manuscript

Answer: I have included it

Comment: Abbreviate nanoparticles as NPs for more clarity.

Answer: I have corrected it in revised manuscript

Comment: Authors have claimed the NPs are monodisperse, which seems to be an exaggerated claim with NPs (Page no. 7; line 267).

Answer: I agree with the reviewer concern regarding the statement in the manuscript. Now we have revised the section as ….

As known, the P.I. is a parameter used to define the particle size distribution of NPs, it is a dimensionless number and ranges from a value smaller than 0.05 are mainly seen with highly monodisperse standards. The values greater than 0.7 showed that the sample has a very broad particle size distribution. The results of our study revealed that the selected sample DNP-7 showed  PDI value lesser than 0.7. Therefore, it can be stated that the kollicoat MAE 30 DP and HPMC-AS based NPs (DNP-7, DNP- 8, DNP-9, DNP-10) prepared by the QESD technique showed homogeneous size distribution (Table 2).

Comment: Results and discussion needed to be supported with suitable references.

Answer: I have updated the result and Discussion section with suitable references.

Comment: Whole manuscript requires to be checked thoroughly and carefully for typographical mistake, incompleteness of sentence, improper usage of grammar etc.

Reply: The manuscript has been revised and the grammer and text error has been recheck.

For example:

Comment: “Cadmium chloride (CdCl2)-induced hepatocellular carcinoma model in male Sprague Dawley rats was used for the antioxidant and hepatoprotective capacity of free and KFP-Nps.”

Answer: The sentence has been corrected.  

Comment “While comparing the protective effects of KFP encapsulated in Kollicoat MAE 30 DP and 25 HPMC-AS Nps was found to be superior to free KFP.”

Answer: The comparison between KFP encapsulated in Kollicoat MAE 30 DP and HPMC-AS Nps and free KFP has been performed and results shown in Table 4, Figure 6.

Comment: In vitro and in vivo should be in italics.

Answer: dear reviewer I have done it italics in revised manuscript.

Comment: In CdCl2, 2 should be in superscript.

Answer: I have subscript 2 in whole manuscript.

Comment: The title itself needed to be revised as: “Formulation and Evaluation of Kaempferol Loaded Nanoparticles Against Experimentally Induced Hepatocellular Carcinoma: In vitro and In Vivo Studies”

Answer: Thanks for your suggestion. I have done a correction in title.

Comment: But at present the formulation of NPs is questionable as results regarding formulation of nanoparticles are not consistent and reproducibility is heavily doubtful with such results.

Answer: The prepared Nps were prepared after prior hit and trial method to select the used polymers. The excipients were found suitable for the delivery system. The prepared Nps were further evaluated for different parameters. There was significant variation in the results were observed. Among them the Nps showed the lowest particle size and high encapsulation efficiency selected for the animal study. The study was performed in triplicate and data shown as mean ± SD.     

Comment: In introduction section (page 2, line 52 – 73), Kaempferol related research work (delivery with novel drug delivery systems), if any available. Should be cited. At present authors have cited research work done on some other drug molecule.

Answer: I have corrected a reference now.

Comment: Page 5, line 202; give no. for subsection “Antioxidant status in Liver”

Answer: I have included a number.

Comment: Mention clearly in all relevant figures/tables representing data, number of repeats of experiment with SD or SEM.

Answer: The figure and Table caption has been revised.

Round 2

Reviewer 3 Report

The NP should be checked by SEM to prove the exact size and morphology whithout the software error and figure 3 should be change to new image with exact scale bar. 

"Figure 3. SEM image of Kaempferol nanoparticle (DNP-7)." should be change "Figure 3. SEM image of Kaempferol nanoparticle (DNP-1)."

Author Response

The NP should be checked by SEM to prove the exact size and morphology whithout the software error and figure 3 should be change to new image with exact scale bar. 

"Figure 3. SEM image of Kaempferol nanoparticle (DNP-7)." should be change "Figure 3. SEM image of Kaempferol nanoparticle (DNP-1)."

ANSWER:- The new image of formulation DNP 7 has been added in manuscript.

Reviewer 4 Report

I would like to thank the authors for the corrections that they did. A lot of their answers were “It was a typo error” even the size values of their nanoparticles (?????). Authors did not show me the welling to improve their manuscript to be published in “Pharmaceutics” Journal. This manuscript still needs more corrections.

1- The new added reference number 12 is not related to Kaempferol as you mentioned in your phrase, it is related to another isoflavonoid. Please correct your phrase or change your reference for the second time. Further, if you add “several studies” in your phrase please add more than “one” reference.

2- In section 3.1, The reference “Brindise MC, Busse MM, Vlachos PP. Density and Viscosity Matched Newtonian and non-Newtonian Blood-Analog Solutions with PDMS Refractive Index. Exp Fluids. 2018 Nov;59(11):173” was repeated two times taking 2 different numbers “16 and 18”. Please correct, it should take only one number.

3- In section 3.1, you have a reference with a number of 178 please correct.

4- In section 3.4, the equation still has the number 2. Please correct

5- In line 161, “The cells were separated from TPVG solution” please add the full name of the abbreviation “TPVG” and Do you mean separated “by”?

6- In line 164, “The cell viabilities of KFP-Np in HepG2 cells damaged by Cd” do you mean “The cell viabilities of KFP-Np-treated HepG2 cells then damaged by Cd”?

7- Lines 165-166, did you treat the cells after 24 h of seeding or at the seeding moment? Please check your methodology and correct these 2 lines accordingly.

8- In line 175 “The animalsf or the study used from” please correct.

9- Line 177 “at room temperature emp” please correct.

10- Please correct the number of section 3.9, 3.10 and 3.12 as they are written as 3.90, 3.101 and 3.123, respectively.

11- Please correct the super- and subscript through the whole manuscript. For example section in 3.9

12- In line 261, “while the other formulations were found in the range of 201-907 nm” please correct to “248 -907 nm”

13- Authors answered to my comment number 30 that the sample is DNP-1. Please provide us with the SEM of DNP-7.

14- I did not understand the authors’s answer to my comment 32 related to Figure 4. Your answer means that you studied the release after 2 h at pH 1.2 and you draw this point in your figure then 1 h at pH 6.8 and you draw this point in your figure?? Please provide us with separated curves for each pH used to show the kinetics of the release. You can find an example here in the reference that you provided me with “Trendafilova, Ivalina & Lazarova, Hristina & Chimshirova, Ralitsa & Trusheva, Boryana & Koseva, Neli & Popova, Margarita. (2021). Novel kaempferol delivery systems based on Mg-containing MCM-41 mesoporous silicas. Journal of Solid State Chemistry. 301. 122323. 10.1016/j.jssc.2021.122323.”

15- In section 4.2.7, “The CdCl2 (20 μM)-exposed HepG2 cells exhibited cell viabilities of 58% and 39% compared to untreated controls (Figure 5)” can the authors explained to me from where they got these values? All the values presented in Figure 5 are above 40% of cell viability. Further, you described in your figure legend that the bar “control” represents the CdCl2 (20 μM)-exposed HepG2 cells which means that we have one value. Another remark please provide us with values ± SD.

16- In line 335, please correct the number of Figure to be” Figure 5”

17- Please provide the missing statistical analyses in vivo (a) Comparing KFP+CdCl2and KFP-NP +CdCl2 to control (b) Comparing KFP+CdCl2to KFP-NP +CdCl2.

18- In line 344, “the supplement of KFP-Np significantly decreased serum levels of ALT, AST, and TBil (p < 0.05, p < 0.01 and p < 0.001)” these 3 parameters in Table 4 have *** which mean p < 0.001. from where the authors got these values. The same comment for the values SOD, CAT and GST in line 350.

19- The quality of Figures 2 should be improved.

20- Scale bar and letters “A, B, C, D” are missing in Figure 5 especially you described your results with letter (e.g. in line 360)

21- Figure 6, included 3 graphs letters as A, B, and C should be included especially you described your results with letter (e.g. in line 378) that does not exist in the figure.

22- In discussion, line 440, “PDI of NPs ranges between the monodispersed particles value from 0.05 to 0.7.” please correct.

Author Response

I would like to thank the authors for the corrections that they did. A lot of their answers were “It was a typo error” even the size values of their nanoparticles (?????). Authors did not show me the welling to improve their manuscript to be published in “Pharmaceutics” Journal. This manuscript still needs more corrections.

1- The new added reference number 12 is not related to Kaempferol as you mentioned in your phrase, it is related to another isoflavonoid. Please correct your phrase or change your reference for the second time. Further, if you add “several studies” in your phrase please add more than “one” reference.

ANSWER:- Dear reviewer, I have replaced a reference 12 with corrected one. And also changed a “several studies” term with “the study”

2- In section 3.1, The reference “Brindise MC, Busse MM, Vlachos PP. Density and Viscosity Matched Newtonian and non-Newtonian Blood-Analog Solutions with PDMS Refractive Index. Exp Fluids. 2018 Nov;59(11):173” was repeated two times taking 2 different numbers “16 and 18”. Please correct, it should take only one number.

ANSWER:- Dear reviewer, I have replaced a reference no 18.

3- In section 3.1, you have a reference with a number of 178 please correct.

ANSWER:-Dear reviewer, it is a typo error.  I have corrected it.

4- In section 3.4, the equation still has the number 2. Please correct

ANSWER:-I have corrected a number of equation

5- In line 161, “The cells were separated from TPVG solution” please add the full name of the abbreviation “TPVG” and Do you mean separated “by”?

ANSWER:- I have added as full form of TPVG “Trypsin Phosphate Versene Glucose (TPVG)” in manuscript and yes its “by”

6- In line 164, “The cell viabilities of KFP-Np in HepG2 cells damaged by Cd” do you mean “The cell viabilities of KFP-Np-treated HepG2 cells then damaged by Cd”?

ANSWER:-  Corrected sentence is “The cell viabilities of KFP-Np in Cd damaged HepG2 cells”. I have corrected a sentence in manuscript

7- Lines 165-166, did you treat the cells after 24 h of seeding or at the seeding moment? Please check your methodology and correct these 2 lines accordingly.

ANSWER:- We have added a various concentration of KFP Nps at the time of seeding in cells.

8- In line 175 “The animals or the study used from” please correct.

ANSWER:- Dear Reviewer, I have corrected a sentence.

9- Line 177 “at room temperature emp” please correct.

ANSWER:- Dear Reviewer, I have corrected a sentence.

10- Please correct the number of section 3.9, 3.10 and 3.12 as they are written as 3.90, 3.101 and 3.123, respectively.

ANSWER:- Dear Reviewer, I have corrected the subheadings.

11- Please correct the super- and subscript through the whole manuscript. For example section in 3.9

ANSWER:- Dear Reviewer, I have corrected.

12- In line 261, “while the other formulations were found in the range of 201-907 nm” please correct to “248 -907 nm”

ANSWER:- Dear Reviewer, I have corrected.

13- Authors answered to my comment number 30 that the sample is DNP-1. Please provide us with the SEM of DNP-7.

ANSWER:- Revised image of SEM image of formulation DNP 7 has been added in revised manuscript.

14- I did not understand the authors’s answer to my comment 32 related to Figure 4. Your answer means that you studied the release after 2 h at pH 1.2 and you draw this point in your figure then 1 h at pH 6.8 and you draw this point in your figure?? Please provide us with separated curves for each pH used to show the kinetics of the release. You can find an example here in the reference that you provided me with “Trendafilova, Ivalina & Lazarova, Hristina & Chimshirova, Ralitsa & Trusheva, Boryana & Koseva, Neli & Popova, Margarita. (2021). Novel kaempferol delivery systems based on Mg-containing MCM-41 mesoporous silicas. Journal of Solid State Chemistry. 301. 122323. 10.1016/j.jssc.2021.122323.”

ANSWER:- The release studies has been reperformed at two different pH and data shown in revised fig and Table.

15- In section 4.2.7, “The CdCl2 (20 μM)-exposed HepG2 cells exhibited cell viabilities of 58% and 39% compared to untreated controls (Figure 5)” can the authors explained to me from where they got these values? All the values presented in Figure 5 are above 40% of cell viability. Further, you described in your figure legend that the bar “control” represents the CdCl2 (20 μM)-exposed HepG2 cells which means that we have one value. Another remark please provide us with values ± SD.

ANSWER:- Dear reviewer, I have corrected a values and provided in values ± SD

16- In line 335, please correct the number of Figure to be” Figure 5”

ANSWER:-  Dear reviewer, I have corrected a number of figure

17- Please provide the missing statistical analyses in vivo (a) Comparing KFP+CdCl2and KFP-NP +CdClto control (b) Comparing KFP+CdCl2to KFP-NP +CdCl2.

ANSWER:- Dear reviewer, we have compared a control group with CdCl2 and then KFP+CdCl2 or KFP-NP +CdCl2 to CdCl2.

18- In line 344, “the supplement of KFP-Np significantly decreased serum levels of ALT, AST, and TBil (p < 0.05, p < 0.01 and p < 0.001)” these 3 parameters in Table 4 have *** which mean p < 0.001. from where the authors got these values. The same comment for the values SOD, CAT and GST in line 350.

ANSWER: Dear reviewer, I have corrected a sentences in revised manuscript.

19- The quality of Figures 2 should be improved.

ANSWER: Dear reviewer, I have improved a figure 2 quality and replaced it.

20- Scale bar and letters “A, B, C, D” are missing in Figure 5 especially you described your results with letter (e.g. in line 360)

ANSWER: Dear reviewer, I have added a scale bar and letters in Figure 6 as per your instructions.

21- Figure 6, included 3 graphs letters as A, B, and C should be included especially you described your results with letter (e.g. in line 378) that does not exist in the figure.

ANSWER: Dear reviewer, I have corrected in Figure 7. Figure 7A-C explained in section 4.2.10.

22- In discussion, line 440, “PDI of NPs ranges between the monodispersed particles value from 0.05 to 0.7.” please correct.

ANSWER:. I have corrected a sentence in manuscript.

Reviewer 6 Report

The manuscript in current form may be accepted.

Author Response

Thanks 

Round 3

Reviewer 4 Report

I would like to thank the authors for the corrections that they did. They addressed some points but I still have some comments to improve this manuscript.

1- Thanks so much for adding the SEM images of DNP-7. I would appreciate if you add the image with higher magnifications. And please correct the name of the sample in section 4.2.3.

2- For the release studies. Please correct your methodology in section 3.6 as you performed the experiment during 12 h. Further, were is the curve at pH 7.4? did you exclude it from the study? Also, in line 342 “The maximum re-341 lease was found 58.98 ± 2.78 (DNP-8), 54.11 ± 3.65 (DNP-7), and 50.22 ± 2.11 (DNP-6).” please check your results as I see a green point “DNP-9” is higher than the orange point “DNP-6”.

3- Concerning the statistical analyses in vivo, I really do not understand why the authors are always refusing to do it. Last time, I had the comment “Now if we do statistical analysis between KFP+CdCl2and KFP-NP +CdCl2 then we have to for all data and it will require to revise a whole manuscript.”. For me it was not an adequate answer at all. Are you considering that these analyses will make you losing time?? If you study therapeutic effect of new formula, you MUST provide the following data (a) Comparing KFP+CdCl2 and KFP-NP +CdCl2 to control (b) Comparing KFP+CdCl2 to KFP-NP +CdCl2. These analyses are important in order to know the difference between free drug and encapsulated one and also if your NPs make the parameter returns to normal levels “as control” or not.

Author Response

I would like to thank the authors for the corrections that they did. They addressed some points but I still have some comments to improve this manuscript.

1- Thanks so much for adding the SEM images of DNP-7. I would appreciate if you add the image with higher magnifications. And please correct the name of the sample in section 4.2.3.

Response: Thank you very much for suggested correction in the manuscript. A magnification (600 DPI) SEM image has been replaced. The DNP-7 has been corrected in section 4.2.3.

2- For the release studies. Please correct your methodology in section 3.6 as you performed the experiment during 12 h. Further, were is the curve at pH 7.4? did you exclude it from the study? Also, in line 342 “The maximum re-341 lease was found 58.98 ± 2.78 (DNP-8), 54.11 ± 3.65 (DNP-7), and 50.22 ± 2.11 (DNP-6).” please check your results as I see a green point “DNP-9” is higher than the orange point “DNP-6”.

Response: The methodology has been revised in section 3.6. There was no significant difference at pH 7.4 was found. Only 1-2 % difference was observed so we have omitted the data at pH 7.4. I agree with the reviewer DNP-9 has slightly higher release than DNP-6. I have corrected in the revised draft. The correction has been highlighted in red colour font.

3- Concerning the statistical analyses in vivo, I really do not understand why the authors are always refusing to do it. Last time, I had the comment “Now if we do statistical analysis between KFP+CdCl2and KFP-NP +CdClthen we have to for all data and it will require to revise a whole manuscript.”. For me it was not an adequate answer at all. Are you considering that these analyses will make you losing time?? If you study therapeutic effect of new formula, you MUST provide the following data (a) Comparing KFP+CdCland KFP-NP +CdClto control (b) Comparing KFP+CdClto KFP-NP +CdCl2. These analyses are important in order to know the difference between free drug and encapsulated one and also if your NPs make the parameter returns to normal levels “as control” or not.

Response: Dear reviewer, I apologize for this mistake happened in last two revisions. I have now done a correction as per your comments in manuscript. I have showed a comparison KFP+CdCland KFP-NP +CdClto control as well as KFP+CdClto KFP-NP +CdCl2.